

**Sources and atmospheric processing of wintertime aerosols**
**in Seoul, Korea: Insights from real-time measurements using**
**a high-resolution aerosol mass spectrometer**
**Hwajin Kim[1,2], Qi Zhang[3,4*], Gwi-Nam Bae[1,2], Jin Young Kim[1,2], Seung Bok Lee[1,2]**
[1] Center for Environment, Health and Welfare Research, Korea Institute of Science and
Technology, Seoul, Korea
[2] Department of Energy and Environmental Engineering, University of Science and Technology,
Daejeon, Korea
[3] Department of Environmental Toxicology, University of California, Davis, CA 95616, USA
[4] Department of Environmental Science and Engineering, Fudan University, Shanghai, China.
*Corresponding author: Qi Zhang
Department of Environmental Toxicology, University of California 1 Shields Avenue, Davis,
California 95616
Phone: (530)-752-5779
Email: dkwzhang@ucdavis.edu
**Abstract**
Highly time-resolved chemical characterization of non-refractory submicrometer particulate
matter (NR-PM$_1$) was conducted in Seoul, the capital and largest metropolis of Korea, using an
Aerodyne high-resolution time-of-flight aerosol mass spectrometer (HR-ToF-AMS). The
measurements were performed during winter, when elevated particulate matter (PM) pollution
events are often observed. This is the first time that detailed real-time aerosol measurement results
are reported from Seoul, Korea, which reveal valuable insights into the sources and atmospheric
processes that contribute to PM pollution in this region.





The average concentration of submicron aerosol ($PM_1$ = NR-$PM_1$ + black carbon (BC)) was
27.5 µg $m^{-3}$, and the total mass was dominated by organics (44%), followed by nitrate (24%) and
sulfate (10%). The average atomic ratios of oxygen-to-carbon (O/C), hydrogen-to-carbon (H/C),
and nitrogen-to-carbon (N/C) of organic aerosol (OA) were 0.37, 1.79, and 0.022, respectively,
which gives that average organic mass-to-carbon (OM/OC) ratio of 1.67. The concentrations (2.6–
90.7 µg $m^{-3}$) and composition of $PM_1$ varied dynamically during the measurement period, due to
the influences of different meteorological conditions, emission sources, and air mass origins. Five
distinct sources of OA were identified via positive matrix factorization (PMF) analysis of the HR-
ToF-AMS data: vehicle emissions represented by a hydrocarbon like OA factor (HOA; O/C =
0.06), cooking activities represented by a cooking OA factor (COA; O/C = 0.15), wood
combustion represented by a biomass burning OA factor (BBOA; O/C = 0.34), and secondary
organic aerosol (SOA) represented by a semi-volatile oxygenated OA factor (SV-OOA; O/C =
0.56) and a low volatility oxygenated OA factor (LV-OOA; O/C = 0.68). On average, primary OA
(POA = HOA + COA + BBOA) accounted for 59% the OA mass whereas SV-OOA and LV-OOA
contributed 15% and 26%, respectively.
Our results indicate that air quality in Seoul during winter is influenced strongly by
secondary aerosol formation with sulfate, nitrate, ammonium, SV-OOA, and LV-OOA together
accounting for 64% of the $PM_1$ mass during this study. However, aerosol sources and composition
were found to be significantly different between clean and polluted periods. During stagnant
periods with low wind speed (WS) and high relative humidity (RH), PM concentration was
generally high (average ± 1σ = 43.6 ± 12.4 µg $m^{-3}$) with enhanced fractions of nitrate (27%) and
SV-OOA (8%), which suggested a strong influence from local production of secondary aerosol.
Low PM loading periods (12.6 ± 7.1 µg $m^{-3}$) tended to occurr under higher WS and lower RH
conditions and appeared to be more strongly influenced by regional air masses, as indicated by
higher mass fractions of sulfate (12%) and LV-OOA (21%) in $PM_1$. Overall, our results indicate
that PM pollutants in urban Korea originate from complex emission sources and atmospheric
processes and that their concentrations and composition are controlled by various factors including
meteorological conditions, local anthropogenic emissions, and upwind sources.





## 1 Introduction

Ambient aerosols can reduce visibility, damage human health, and influence climate change directly by absorbing and reflecting solar radiation, and indirectly by modifying cloud formation and properties (IPCC, 2013;Pope III and Dockery, 2006;Pöschl, 2005). Elevated particulate matter (PM) pollution in urban areas are known to arise due to anthropogenic emissions and stagnant meteorological conditions (Cao et al., 2012;Guo et al., 2014;Sun et al., 2014).

Seoul is one of the most populated and developed city in Korea and is facing a severe PM pollution problem. The annual average concentration of $PM_{2.5}$ in Seoul decreased from 28.5 µg m$^{-3}$ in 2005 to 22.5 µg m$^{-3}$ in 2012 since the enactment of the "Special Act on Seoul Metropolitan Air Quality Improvement" in 2005, which has led to emission reduction from diesel vehicles and fugitive dust emissions on road and open area (Kang et al., 2016;KOSAE, 2009). However, the amount of reduction was not dramatic and the annual average $PM_{2.5}$ concentration increased again in 2013 to 24.8 µg m$^{-3}$. These values far exceeded the $PM_{2.5}$ annual standards set by the USA (15 µg m$^{-3}$) and the WHO (10 µg m$^{-3}$). Due to growing concerns over the adverse effects of atmospheric PM, the South Korean government established $PM_{2.5}$ standards in 2015: 25 $\mu$g m$^{-3}$ for annual average and 50 $\mu$g m$^{-3}$ for 24-hr average (NIER, 2014).

Mitigating air pollution in Seoul remains a great challenge because it remains unclear which emission sources and atmospheric process are responsible for the problem (Harrison and Yin, 2000), despite numerous studies that have focused on these issues worldwide (e.g., Ervens et al., 2011;Gelencsér et al., 2007;Jimenez et al., 2009;Ng et al., 2010;Young et al., 2016;Zhang et al., 2007a). A main reason is the complexity of ambient aerosols, which have a range of chemical compositions and originate from a wide range of sources and atmospheric processes (Seinfeld and Pandis, 2006). Furthermore, several recent studies investigate the impacts of meteorology on air pollution, as well (Ding et al., 2013;Ding et al., 2016;Petaja et al., 2016). For example, high PM concentration was found to happen more frequently during winter due to a combination of factors, including elevated emissions from primary sources for heating, lower boundary layer (BL) height, and stagnant meteorological conditions, which favor the accumulation of PM and secondary aerosol precursors. In addition, cold weather in winter promotes the gas-to-particle partitioning of semi-volatile species such as nitrate. In addition, since Seoul is located downwind from the Asian Continent, where concentrations of ambient aerosols have increased significantly in recent years (Huang et al., 2014;Liu et al., 2013;Quan et al., 2011), long-range transport of pollutants from



upwind polluted areas often influences air quality in Seoul, especially during winter (Kim et al.,
2014). As a result, both local and regional sources contribute to high concentrations of PM, both
separately and in combination (Heo et al., 2009), making air quality control more difficult.
Therefore, understanding the sources and processes responsible for PM composition and
concentration is critical for the public to recognize regional haze events and to enact effective PM
reduction strategies for Korea as well as for the broader northern pan-Eurasian region (Liu et al.,
2013). Also, from a global point of view, studies of atmospheric PM pollution may contribute to
a better understanding of the Earth system science and enacting efficient climate policy (Kulmala
et al., 2015).
Most previous aerosol studies in Korea were based on offline sampling which involves filter
collection followed by laboratory analyses. However, a main drawback of this method is low time-
resolution which limits our ability to perform detail investigations on the atmospheric evolution
processes. Furthermore, organic aerosol (OA) analysis conducted in Korea has so far focused on
molecular level, which led to the identification of a small portion of the OA mass (Choi et al.,
2012;Choi et al., 2016). Aerosol mass spectrometers (AMS) have been widely used in recent years
because they allow for chemical speciation, sizing and the mass detection of submicron non-
refractory submicrometer particulate matter (NR-PM$_1$), with a high time resolution (Canagaratna
et al., 2007). Furthermore, using multivariate analysis methods (Zhang et al., 2011), the OA was
decomposed into several sources such as hydrocarbon-like OA (HOA), biomass burning OA
(BBOA), cooking-related OA (COA), coal combustion OA (CCOA), and oxygenated OA (OOA).
In addition, the OOA factor has been further separated according to volatility, e.g., low volatility
fraction (LV-OOA) and a semi-volatile fraction (SV-OOA) or oxidation, e.g., more oxidized (MO-
OOA) and less oxidized (LO-OOA) (e.g., Lanz et al., 2007;Parworth et al., 2015;Setyan et al.,
2012;Aiken et al., 2008;Huang et al., 2011;Mohr et al., 2012). These studies have identified the
sources, constituents and the evolution process of PM which is critical for establishing efficient
control strategies and model representations (Ulbrich et al., 2009;Zhang et al., 2011).
Many AMS studies have been conducted in East Asia including China, Hong Kong, and Japan,
but a detailed PM characterization using AMS has not yet been reported in Seoul, Korea (Li et al.,
2015 and references therein). In this study, we deployed a high-resolution time-of-flight aerosol
mass spectrometer (HR-ToF-AMS), manufactured by Aerodyne Research Inc. (Billerica, MA,
USA), in Seoul for 6 weeks  (from December 5, 2015 to January 21, 2016) to characterize





wintertime NR-PM$_1$ in this urban area. We particularly focus on aerosol properties, sources, and
processes in winter, when the weather was generally cold and relatively dry during the day but was
humid during the night. Our goal was to use detailed information obtained from in situ
measurements to facilitate a fundamental understanding of the formation processes and emission
sources of atmospheric aerosols in Seoul, which may improve our understandings on what factors
and how they influences urban air quality. This information will eventually allow for the design
of better pollution abatement strategies and improved parameterizations in air quality models. Here,
we report: (1) the mass concentrations, size distributions, chemical composition, and temporal and
diurnal variations of PM$_1$ species; (2) the characteristics and dynamic variations of OA sources
and processes using PMF; and (3) the characteristics, sources, and impact factors of the PM$_1$
composition and OA components on polluted and unpolluted days.
**2    Experimental Methods**
**2.1   Sampling site description**
Seoul, the capital of Korea, is located in the central-west of the Korean Peninsula surrounded by
the Yellow Sea. Korea is generally under the influence of a prevailing north-westerly wind
bringing air masses originating from mainland China during winter (Fig. 1). In this study, real-
time measurements of particle composition and size distribution were conducted at the Korea
Institute of Science and Technology (KIST) located in the northeast of Seoul (37.60N, 127.05E),
7 km from the city center. The sampling was conducted on the 5$^{th}$ floor of one of the KIST
buildings (60 m above sea level). As shown in Fig. 1d, KIST is located ~ 400 m from a busy
highway and is surrounded by a residential area, forest and a commercial area, suggesting that the
air quality at the site tends to be influenced by abundant anthropogenic and primary sources. There
are no major industrial or wood burning sources in the vicinity of KIST. However, in the west and
west south part of Seoul, ~ 35 km away from the measurement site (Fig. 1b), there are a number
of industrial facilities that are significant anthropogenic sources. There are also sporadic
agricultural areas surrounding the city of Seoul (e.g., Gyeonggi), where open biomass burning
often times occurs during winter. A wide range of meteorological and air quality data were also
collected from a separate supersite located at 500 m away from the northwest of the KIST.



## 2.2 Measurements

The NR-PM$_1$ chemical components including sulfate, nitrate, ammonium, chloride, and organics as well as their size distributions were measured by an Aerodyne HR-ToF-AMS (DeCarlo et al., 2006) at a time resolution of 2.5 min. The black carbon (BC) concentration was measured with a multi angle absorption photometer (MAAP; Thermo Fisher Scientific, Waltham, MA, USA). Both instruments sampled downstream of a PM$_{2.5}$ cyclone (URG Corp.; Chapel Hill, NC, USA). The number size distributions of aerosols with mobility sizes between 20.9–947.5 nm were measured by a scanning mobility particle sizer (SMPS 3080; TSI Inc., St Paul, MN, USA). The hourly concentration of trace gas (e.g., CO, O$_3$, NO$_2$ and SO$_2$) measured at Gireum site (37.61N, 127.03E) were acquired from the Korea Environment Cooperation (KECO) (http://www.airkorea.or.kr). Meteorological measurement data such as ambient temperature, relative humidity (RH), wind speed and wind direction were obtained from the nearby Jungreung site (37.61N, 127.00E) maintained by the Korea meteorological administration (http://www.kma.go.kr). The data reported in this paper are in local time, which is Korea Standard Time (KST) and is 9 h earlier than the Universal Coordinated Time (UTC).

In this study, the HR-ToF-AMS was operated in the standard configuration and obtained mass spectra (MS) and efficient particle time of flight (ePToF) data. Furthermore, the HR-ToF-AMS was operated under the 'V' and 'W' modes, where high sensitivity but low mass resolution was achieved in 'V' mode, and low sensitivity, but high mass resolution was achieved in 'W' mode. Ionization efficiency (IE) and particle sizing calibrations were performed following standard protocols (Canagaratna et al., 2007) before, during, and after the measurement period.

## 2.3 AMS data analysis

### 2.3.1 Basic HR-ToF-AMS data analysis

HR-ToF-AMS data were processed and analyzed using the standard toolkit (SeQUential Igor data RetRiEval (SQUIRREL; ver. 1.57I), and PIKA (ver. 1.16H; available for download at http://cires.colorado.edu/jimenez-group/ToFAMSResources/ToFSoftware/index.html)) within Igor Pro (Wavemetrics, Lake Oswego, OR, USA). Details on the data processing procedures have been described in previous studies (Aiken et al., 2008;Aiken et al., 2009;Allan et al., 2004). Briefly, the standard fragmentation table described by Allan et al. (2004) was used, with some small modifications to process the raw MS. The modifications were based on data from four



filtered air measurements. This allowed measurements of the background from the gas-phase
signal, which needed to be removed from the particle-phase measurement. Adjustments were made
to the measured $CO_2^+$ ($m/z = 44$) signal to remove the contributions from gas phase $CO_2$ as well
as the $^{16}O^+$ to $^{14}N^+$ ratio for air signals at $m/z = 29$ based on measurements of particle-free ambient
air. Relative ionization efficiencies (RIE) of 1.1, 1.075, and 3.875 were used for nitrate, sulfate,
and ammonium, respectively, based on values determined from calibrations using pure $NH_4NO_3$
and $(NH_4)_2SO_4$ aerosols. A composition-dependent collection efficiency (CDCE) was applied to
the data based on an algorithm by Middlebrook et al. (2012). Nitrate was observed to be an
important component of NR-$PM_1$ during this study (24%), although the campaign average ($\pm 1\sigma$)
CE was $0.5 \pm 0.02$.
The quantification of NR-$PM_1$ species was validated through comparisons between the total $PM_1$
mass ($PM_1$ = NR-$PM_1$ + BC) and the apparent particle volume measured by the SMPS (Fig. 2e,
Fig. S1a). As shown in Fig. S1a, the SMPS-measured particle volume correlated strongly with the
AMS measured total mass ($R^2 = 0.9$). From this strong correlation, the inter-comparison of AMS
mass versus SMPS volume yielded a slope of 1.22, which was lower than the average particle
density  of 1.46 estimated   using the measured chemical composition in this study (Zhang et al.,
2005b) (Fig.3c and Fig. S1b). Note that the average organic aerosol density was estimated to be
1.12 based on elemental ratios (Kuwata et al., 2012). The evolution pattern of the AMS total mass-
based size distribution also compared well with the volume-based size distribution from SMPS
measurements throughout the day (Fig. S1c,d). The detection limits of the main chemical
components in AMS are listed in Table 1, and are typically far lower than the observed
concentrations. All the reported mass concentrations from AMS in this study are based on ambient
conditions.
The elemental ratios between oxygen, carbon, hydrogen, and nitrogen as well as the organic mass
to carbon ratio (OM/OC) of OA, were determined from an analysis of the W mode high resolution
mass spectra (HRMS) data, following the method recently reported by Canagaratna et al. (2015).
The elemental ratios calculated using the Aiken-Ambient method (Aiken et al., 2008) are detailed
in Table S1 along with the ratios calculated using the Canagaratna method for comparison. Unless
otherwise indicated, the O/C, H/C, and OM/OC ratios stated in this paper from other studies have
been calculated using the updated elemental analysis method (Canagaratna et al., 2015).
***2.3.2  Positive Matrix Factorization (PMF) Analysis***





The HRMS of organic aerosol were analyzed using PMF. The analysis was performed using the
PMF2 algorithm in robust mode (Paatero and Tapper, 1994), with the PMF Evaluation Toolkit
(PET ver 2.05) (Ulbrich et al., 2009), which was downloaded from
http://cires1.colorado.edu/jimenez-group/wiki/index.php/PMF-
AMS_Analysis_Guide#PMF_Evaluation_Tool_Software. The data and error matrices were
prepared according to the protocol described by Ulbrich et al. (2009) and outlined in Table 1 of
Zhang et al. (2011). In brief, a minimum error value was applied to the error matrix and each ion
was assessed and treated according to their signal-to-noise ratio (SNR). Ions with an average SNR
of less than 0.2 were removed, and those with SNR between 0.2 and 2 were downweighted by
increasing their errors by a factor of two. Further, ions related to $m/z$ 44 (i.e., $CO_2^+$, $CO^+$, $H_2O^+$,
$HO^+$, and $O^+$) were also downweighted to avoid overestimating the importance of $CO_2^+$. Isotopes
were removed from the matrices because their signals were scaled to their parent ions rather than
being measured directly. After these treatments, the resulting matrix consisted of 286 ions between
$m/z$'s 12 and 120.
PMF was applied to the data and the number of factors ($p$) in the solution was explored from one
up to nine. Although the PMF algorithm was able to provide a number of mathematically sound
solutions, in order to obtain physically meaningful results, several criteria were used to evaluate
and select the appropriate number of factors from the model. The recommendations outlined in
Zhang et al. (2011) including an investigation of the key diagnostic plots, mass spectral signatures,
diurnal profiles, and correlations with external tracers, were followed to assess the results and
determine the number of factors.
The rotational ambiguity of each of the solution sets was explored by varying the fPeak parameter
from -1 to 1, with an increment of 0.1. The five factor solution with fPeak = 0 ($Q/Q_{exp}$ = 2.56), was
selected for further analyses because it satisfied the above criteria, including a good separation of
the temporal and mass spectral variations of the five factors. A summary of the key diagnostics is
presented in Fig. S2 in the Supplement. The five factor solution was found to be very stable as the
mass fraction of each of the factors remained relatively constant between fPeaks -0.4 and +0.2
(Fig. S2c). Fig. S3 shows the mass spectra and the time series of the four and six factor solutions;
where factor 1 and factor 2 in the four factor solution set could be identified as low volatility OA
(LV-OOA) and semi-volatile OA (SV-OOA) based on the O/C ratio, but were both highly oxidized,
with a higher fraction of $m/z$ 44 than $m/z$ 43. Moreover, all factors in this solution set showed the





signature of biomass burning OA (BBOA) at *m/z* 60 and *m/z* 73, indicating that more factors may
be needed to resolve the mixed factors. In contrast, the temporal variations from the six factor
solution were similar to those from the five factor solution set but showed indications of factor
splitting. For example, two very similar BBOA-like factors (factors 3 and 4) were identified in the
six factor solution, where the main difference was that factor 4 had a higher N/C ratio than factor
3. Although different types of BBOA sources are possible, we did not have external evidence to
validate the separation. Furthermore, given the fact that having only one BBOA factor (i.e., the
five factor solution set) did not influence the separation of the other factors, it was not necessary
to go for higher number of factors. Consequently, the five factor solution, which resolved HOA,
COA, BBOA and two types of OOA, was chosen as it appears to best represent OA sources and
processes in Seoul during winter.
**2.4   Backtrajectory and Bivariate conditional probability function analysis**
In this study, 96-h back trajectories were calculated every hour using version 4.8 of the Hybrid
Single-Particle Lagrangian Integrated Trajectory (HYSPLIT) model (Draxler, 2012;Draxler, 1997)
for the sampling periods from December 5, 2015 to January 21, 2016. The ending location for the
trajectories was the KIST (latitude: 37.60N; longitude: 127.05E) at an elevation of 500m. To
identify the pollutant characteristics in different predominant transport patterns, cluster analysis
was performed on the trajectories using the software HYSPLIT4 and 4 clustered were identified
according to their similarity in a spatial distribution.
In addition, conditional probability function (CPF) (Kim et al., 2003) was performed to estimate
the local sources and their impacts on $PM_1$ composition and individual organic aerosol sources
from PMF analysis, using wind directions coupled with the concentration time series of each
species. The CPF plots represent the probability that a specific compound or source is located in
certain wind direction, assisting to find the local point source. The directional origin of regionally
transported sources may not be consistent with the local surface wind data used for the CPF plots
due to the tophography of the region (Heo et al., 2009).
**3   Results and discussion**
**3.1   Overall characteristics**
*3.1.1 Temporal variations of $PM_1$ composition and chemical properties*





The overall characteristics and temporal variations of wintertime $PM_1$ in Seoul, including mass
and volume concentrations, and size distributions are shown in Fig. 2, along with the time series
of gaseous pollutants, e.g., CO, $SO_2$, and $O_x$ ($O_x = O_3 + NO_2$), and meteorological conditions (RH,
temperature, wind direction, wind speed). From Dec. 6, 2015 to Jan. 21, 2016, the average
concentration of $PM_1$ (= NR-$PM_1$ + BC) was 27.5 µg m$^{-3}$, ranging from 2.6 to 90 µg m$^{-3}$. Assuming
that $PM_1$ represents approximately 80% of $PM_{2.5}$ mass (Lim et al., 2012), we found that 29% of
the measurement days (i.e., 14 days) violated the NIER's daily $PM_{2.5}$ standards (50 µg m$^{-3}$) and
58% of the days (28 days) violated the WHO's daily standard (25 µg m$^{-3}$). Although severe haze
with $PM_1$ concentration close to 90 µg m$^{-3}$ was observed frequently, the averaged mass
concentration of $PM_1$ (27.5 µg m$^{-3}$) was still moderate because of the frequent occurrence of clean
periods in winter. The average concentration of $PM_1$ measured in Korea during this study was
similar to or slightly lower than those measured during wintertime in several urban areas in China,
including Shanghai (Huang et al., 2012), Shenzhen (He et al., 2011), Lanzhou (Xu et al., 2016),
and Hong Kong (Li et al., 2015), but was much lower than in Beijing where the wintertime mass
concentrations of $PM_1$ were found to be 7-10 times higher than in Seoul  (Sun et al., 2014).
The large variations in $PM_1$ mass concentrations (2.6 to 90 µg m$^{-3}$; Fig. 2e) and other pollutants
(Fig. 2c), such as CO (0.3 to 2.3 ppm), $O_3$ (3 to 46 ppb), and $NO_2$ (8 to 98 ppb), reflected that air
quality in Seoul is influenced by dynamic changes in emission sources, atmospheric processes, as
well as meteorological conditions. In addition, new particle formation events were observed during
this study and they showed characteristics of a sharp increase of ultrafine particle number
concentration and subsequent growth of these particles in size (Fig.2i). This finding is the first
time in urban area of Korea, although frequent (~7.4 % of the measurement days) observations of
new particle formation and growth events were reported from the 3 year continuous measurements
using SMPS at Gosan station (33.17N, 126.10E) which is pristine rural area but downwind site of
Asian Continent facing yellow Sea (Kim et al., 2013).
Based on the variations of $PM_1$ concentrations and meteorological conditions (Fig. 2), we divided
the whole study into two typical periods: (1) high loading period (daily $PM_1$ > 30 µg m$^{-3}$) and (2)
low loading period (daily $PM_1$ < 14 µg m$^{-3}$). 30 and 14 µg m$^{-3}$ correspond to the middle values of
the $PM_1$ concentrations estimated based on the NIER and WHO daily and yearly $PM_{2.5}$ standards,
respectively. As shown in Figs. 2 and 3, high and low loading periods usually alternated during
winter in Seoul. Comparisons between the high and low loading periods can indicate how different





sources and atmospheric processes influence air quality in this region. Details on the periodic
variations of air quality in Seoul are discussed at section 3.4.
On average, OA was the largest component of the $PM_1$, accounting for 44% of the total mass,
followed by nitrate (24%), sulfate (10%), ammonium (12%), BC (9%) and chloride (1%) (Fig. 3a,
4a and Table 1). POA (= HOA + COA + BBOA) and SOA (= SV-OOA + LV-OOA) accounted
for 59 and 41%, respectively, of OA mass (section 3.3). Consequently, ~ 36% of $PM_1$ was consisted
of primary materials (POA + BC), with the remainder (64%) being secondary species ($NO_3^-$ +
$SO_4^{2-}$ + $NH_4^+$ + SOA), indicating that the aerosol pollution problem in Seoul during wintertime
was more strongly influenced by secondary aerosol formation processes. Details on the sources
and processes that led to severe air quality are discussed in section 3.4.
The molar equivalent ratios of total inorganic anions to cation for NR-$PM_1$ (= ($SO_4^{2-}$/48 + $NO_3^-$/62
+ $Cl^-$/35.5) / ($NH_4^+$/18)) were close to 1; thus, submicron aerosols were mostly neutralized in the
forms of ammonium salts, such as $NH_4NO_3$, $(NH_4)_2SO_4$, and $NH_4Cl$ (Zhang et al., 2007b) (Figs.
3d and 4b). Possible sources of ammonium in Seoul include on-road vehicle emissions, neutralizer
using in industry, and agricultural emissions at the outskirt of Seoul. Note that particles appeared
to have "excess" $NH_4^+$ at high organic aerosol loadings (Fig. 4e), probably due to the presence of
carboxylate ions, such as formate, acetate, and oxalate, which were not counted in the calculation
of ion balance. Further investigations of this issue might be necessary in the future.
***3.1.2 Diurnal patterns of $PM_1$ composition***
Diurnal patterns can provide insights into aerosol sources and formation processes. In this study,
the daily variations in concentrations of aerosol species show distinctively different patterns,
indicating that the sources and formation processes of PM pollutants in Seoul were diverse and
complex. First, secondary inorganic species, such as sulfate, nitrate, and ammonium, all displayed
different diurnal profiles. In the case of nitrate (Fig. 5c), the daily variations started to increase at
~ 07:00, peaked at midday (10:00-12:00), and then slowly decreased until 17:00. Previous studies
(Brown et al., 2006;Lurmann et al., 2006;Sun et al., 2012;Young et al., 2016) have attributed this
type of daytime peak shortly after sunrise to the mixing-down of secondary aerosols formed at
night in a residual layer aloft. Later, the photochemical formation of nitrate from $NO_x$, emitted
from rush hour traffic, contributed to an increase in nitrate concentration during daytime, a feature
of regionally generated secondary inorganic species. The decreasing trend after noon is likely due


to the evaporative loss of semi-volatile species at higher air temperatures as well as the dilution
effects due to the enhanced BL height in the afternoon. Significant amount of nitrate can also be
formed through nighttime chemistry, as indicated by the high fractional contribution during night
(Fig. 5f). Given that somewhat high concentrations of $O_3$ (~12.0 ppb) and $NO_2$ (~41.7 ppb) were
observed during night time (18:00 – 6:00), nitrate formation from $N_2O_5$ hydrolysis possibly
occurred.
Unlike nitrate, sulfate concentration was elevated during nighttime, showing a trend that started to
increase from late afternoon (~16:00), peaked at around 10:00 on the following day, and then
gradually decreased afterwards to reach a minimum value at 16:00 (Fig.5b). The overnight increase
starting in the late afternoon appeared to be associated with enhanced gas-to-particle partitioning
of $SO_2$ and aqueous phase processing facilitated by the relatively low temperature and high RH at
night (Fig. 6). Indeed, $fSO_4$ (= the ratio in sulfate ($SO_4^{2-}$) to $SO_x$ ($SO_4^{2-}$ + $SO_2$) based on sulfur
contents (Kaneyasu et al., 1995)) increased during night time and showed relatively good
correlation with RH ($R^2$ = 0.59, Fig. 6a). However, $fSO_4$ started to decrease at ~6:00 (Fig. 6c) but
sulfate concentration continued to increase till 10:00 (Fig. 6b). This increase of sulfate
concentration appeared be due to a similar reason as that of nitrate – mixing with the higher
concentration of sulfate in the upper residual layer formed at night. The residual layer was also
enriched of $SO_2$ (Fig. 6b), for which nighttime transport of air mass from industrial facilities
located on the west and southwest outskirts of Seoul (Fig. 1b) might be responsible. However the
bivariate polar plots (Fig. 7) indicate that high $SO_2$ concentration tended to occur under high speed
wind from the south and southeast directions, which shifted relative to the locations of the $SO_2$
point sources (Fig. 1b). The reason might be geographical since the position (North) of the Bukhan
Mountain block the wind and rather promotes the circulation of air masses. Similar trends were
observed in a previous study (Heo et al., 2009).
The decrease of both sulfate and $SO_2$ concentrations between 10:00 and 17:00 (Fig. 6b) could to
be due to the dilution effect from rising BL height. Since $fSO_4$ showed only minor increases
between 12:00 and 17:00 when RH was low (Fig. 6c), gas phase photochemical production of
sulfate during daytime was unlikely to be an important process. This observation, together with
the nighttime increase of sulfate associated with high RH, suggests that aqueous phase processing
is an important driver for sulfate formation in Seoul during wintertime. This conclusion is
consistent with previous studies which found that aqueous phase reactions were an important



pathway for sulfate formation in the atmosphere under humid conditions (Ervens et al., 2011;Ge
et al., 2012b).
In this study, as mentioned in the earlier section, nitrate, sulfate and chloride in $PM_1$ appeared to
be fully neutralized by ammonium, indicating that the inorganic species were mainly present in
the forms of $NH_4NO_3$, $(NH_4)_2SO_4$ and $NH_4Cl$. Since nitrate was more abundant compared to
sulfate and chloride (Fig. 4a), the ammonium diurnal pattern was similar to that of nitrate. However,
a gradual increase of ammonium during nighttime was observed, likely due to the enhancement of
the sulfate concentration. Chloride accounted for a small fraction of the $PM_1$ mass and displayed
a morning rush hour peak, suggesting that the source of chloride is probably local, such as vehicle
emissions. Indeed, the polar plot of chloride in Fig. 7 does show the feature of local source since
high concentration occurred mostly at slow wind speed. Further, chloride showed a similar diurnal
pattern as HOA (Section 3.3, Fig. 10) and good correlation with HOA (r =0.8; Table 2).
In contrast to ammonium and nitrate, OA concentration tended to remain high overnight and
started to increase in the morning with a maximum usually occurring at 8:00 (Fig. 5a). This
morning increase was likely the outcome of shallow BL coupled with enhanced primary emissions
from rush-hour traffic. Further discussions on this are given in section 3.3.
**3.2   Size distributions of the main components of $PM_1$**
The average mass-based size distributions of AMS species in terms of vacuum aerodynamic
diameter ($D_{va}$, (DeCarlo et al., 2004)) are shown in Fig. 4c. Nitrate and sulfate had relatively
different size distribution profiles with the mode of sulfate being about 100 nm larger than the
mode of nitrate. A possible reason for this difference was likely that nitrate was mainly formed
through photochemical reactions whereas sulfate was more likely to be formed by aqueous-phase
reactions during this study. Another reason was that sulfate was overall more aged than nitrate. As
discussed in previous sections, sulfate in Seoul was likely to be transported from regional sources
during nighttime whereas nitrate was apparently formed mostly locally. The daily variations in the
nitrate and sulfate size distributions (Figs. 5b, c, Fig. S4) further support the different formation
processes for these two compounds. The size distributions of sulfate showed a prevalent droplet
accumulation mode ($D_{va}$ = 500 nm) that stayed fairly constant in concentrations and size
distribution (Fig. S5b). The size distributions of nitrate stayed relatively constant throughout the
day as well, but changed significantly in concentrations (Fig. S4a). In addition, nitrate size
distribution was quite broad (Figs. 4c,d), similar to observations in various urban locations. (e.g.,



Sun et al., 2009;Sun et al., 2011a;Drewnick et al., 2004;Salcedo et al., 2006;Weimer et al.,
2006;Young et al., 2016;Zhang et al., 2005b)
The average size distribution of OA was wider than those of the inorganic species, peaking at a
$D_{va}$ of ~ 400 nm (Fig. 4c). The OA size distribution varied as a function of the time of day (Fig.
5a), with a broader profile extending to $D_{va} < 100$ nm observed during the morning rush hour and
night time when primary emission are dominant. The wide size distribution of organics reflected
the contribution made by both primary and secondary aerosols, i.e., the fine mode from primary
aerosols and the accumulation mode from secondary formation. Similar observations were
reported from China, (e.g., (Huang et al., 2010;Sun et al., 2010)) and some urban areas in North
America (Aiken et al., 2009;Alfarra et al., 2004;Drewnick et al., 2004;Ge et al., 2012b;Zhang et
al., 2005b) and Europe (Allan et al., 2003;Dall'Osto et al., 2013).
**3.3 OA characteristics and source apportionment**
*3.3.1 Bulk composition and elemental ratios of OA*
Atmospheric OA are composed of complex materials that originate from different sources and
have undergone different atmospheric processes. Understanding the chemical composition and
sources of OA is important for understanding the impacts of these aerosols.
Overall, OA from Seoul during wintertime was found to be composed of approximately 71%
carbon, 18% oxygen, 9% hydrogen, and 2% nitrogen (Fig. 4e). The average carbon-normalized
molecular formula of OA was $C_1H_{1.8}O_{0.37}N_{0.022}S_{0.0009}$, yielding an average organic mass-to-carbon
ratio (OM/OC) of 1.67. The largest component of the OA mass spectral signal was found to be the
$C_xH_y^+$ ion family (57%, Fig. 4e), followed by the $C_xH_yO_1^+$ (25%) and $C_xH_yO_2^+$ (11%) ion families,
with smaller contributions from the $C_xH_yN_p^+$ (4%), $C_xH_yN_pO_z^+$ (2%), and $H_yO_1^+$ (1%) ion families.
The largest peak in the average OA spectrum was at $m/z = 43$ (Fig. 4f), accounting for 8% of the
total OA signal with a composition of $C_2H_3O^+$ (49%), $C_3H_7^+$ (49%), $CHON^+$ (1%), and $C_2H_5N^+$
(1%). The second largest peak (6% of the total OA signal) in the average OA spectrum was at $m/z$
= 44, which was dominated by the $CO_2^+$ ion (86.7%) followed by $C_2H_4O^+$ (7.8 %), $C_2H_6N^+$ (2.7%),
$CH_2NO^+$ (2.6%) and $C_3H_8^+$ (0.2%). The peak at $m/z = 60$ was composed almost entirely of $C_2H_4O_2^+$
(94%) and 88% of the peak at $m/z = 73$ was composed of $C_3H_5O_2^+$, both of which are the tracers
of wood burning (Aiken et al., 2008;Alfarra et al., 2007). The peak at $m/z = 57$, which is used as a
tracer for hydrocarbon like organics from vehicle emissions (Zhang et al., 2005a) accounted for





4% of the total OA signal, and was composed predominantly of $C_4H_9^+$ (75%) and $C_3H_5O^+$ (22%)
in this study.
The time series of the H/C, O/C, N/C and S/C ratios of OA are shown in Figs. 3e and f. The O/C
ratio of an OA indicates its average oxidation level and more aged and oxidized organics tend to
have higher O/C ratios (Aiken et al., 2008;Jimenez et al., 2009). The O/C varied substantially
during this study ranging from 0.05 to 0.63 and the average O/C ratio was 0.37±0.09. The average
H/C ratio was 1.79±0.07 (1.60 – 2.29) and the average OM/OC was 1.67±0.12 (1.26 – 2.03). These
values, which were calculated using the updated elemental analysis method (Canagaratna et al.,
2015), are within the range of revised values observed at other urban locations (Canagaratna et al.,
2015 and references within). Upon examining the diurnal patterns, we also found that the O/C and
OM/OC ratios started to increase in the morning and peaked in the afternoon (Figs. 8a and e). The
lowest value of both parameters occurred at around 8:00, due to enhanced vehicle emissions during
morning rush hours coupled with low BL height, which was also evident in the diurnal profile of
the H/C ratio (Fig. 8b). However, during daytime (8:00 – 16:00), O/C ratio generally increased
and H/C ratio decreased suggesting that SOA production or mixing with more aged aerosol from
regional sources was important during daytime and outweighed the emissions POA. The decrease
of the OM/OC and O/C ratios as well as the increase of H/C ratio in the late evening (19:00−
20:00) were consistent with an enhancement of POA emissions during the evening rush hour and
dinner time.
Although organic ions containing nitrogen and sulfur had relatively low abundance (average N/C
= 0.018; average S/C = 0.001), both nitrogen-to-carbon (N/C) and sulfur-to-carbon (S/C) ratios
showed distinct diurnal profiles where N/C enhanced during daytime (8:00 – 16:00) and increased
again from late evening (19:00), suggesting that particulate nitrogen-containing compounds in
Seoul were probably from both primary emissions and secondary formation. However, S/C
showed relatively strong enhancement during daytime (8:00 – 16:00), suggesting that sulfur
containing organics were mainly formed by secondary processes during daytime. This is further
confirmed by considering the correlation between different OA factors vs. AMS spectral ions;
nitrogen containing ions had good correlation both with POA and SOA factors whereas sulfur
containing ions had good correlation only with OOAs (Fig. S5; see section 3.3.2).
***3.3.2 Organic aerosol source apportionment and characteristics of OA factors***





Separation of distinct organic aerosol sources can be achieved through the application of
multivariate models, such as PMF (Lanz et al., 2007;Ulbrich et al., 2009;Zhang et al., 2011). In
this study, five OA factors were determined, consisting of three POA factors (HOA, COA and
BBOA) and two SOA factors (LV-OOA and SV-OOA). The O/C ratios for the factors were: LV-
OOA = 0.58; SV-OOA = 0.47; BBOA = 0.22; COA = 0.14; and HOA = 0.15. The elemental ratios
of the factors were estimated using the updated method reported by Canagaratna et al. (2015). A
comparison of the O/C and H/C ratio of each PMF factor, as determined by the methods of Aiken
et al. (2008) and Canagaratna et al. (2015), can be found in Table S1. An overview of the chemical
composition and temporal variations of the five factors is shown in Figs. 9, 10 and Fig. S7. The
five factors made similar contributions to total OA mass, with LV-OOA (26%) representing the
largest fraction of the OA mass and the smallest faction accounted for by SV-OOA (15%). BBOA,
COA and HOA accounted for 23, 20 and 16% of the total OA mass, respectively. Together,
primary components on average accounted for 59% of the total OA mass and SOA accounted for
41% (Fig. 10k). The chemical composition and temporal variations of each factor are discussed in
detail below.
*3.3.2.1 Hydrocarbon-like OA (HOA)*
Alkyl fragments ($C_nH_{2n+1}^+$ and $C_nH_{2n-1}^+$) made a substantial contribution to the HOA factor, with
major peaks at *m/z*'s 41, 43, 55, and 57 which were mostly composed of $C_3H_5^+$, $C_3H_7^+$, $C_4H_7^+$, and
$C_4H_9^+$ ions, respectively (Fig. 10a). These major peaks and the overall picket fence fragmentation
pattern resulting from the $C_nH_{2n+1}^+$ ions are typical features of the HOA spectra reported in other
studies and are due to the association of these aerosols with fossil fuel combustion (e.g.,Alfarra et
al., 2007;Lanz et al., 2008;Sun et al., 2011b;Zhang et al., 2005a;Huang et al., 2010;Morgan et al.,
2010;Ng et al., 2011). In addition, strong correlations were observed between the time series of
HOA and the $C_nH_{2n+1}^+$ and $C_nH_{2n-1}^+$ ions, e.g., $C_3H_7^+$ (r = 0.91), $C_4H_7^+$ (r = 0.85), $C_4H_9^+$ (r = 0.95),
and $C_5H_{11}^+$ (r = 0.96) (Fig. 9a and Table 2). Due to the dominance of chemically reduced
hydrocarbon species, the O/C ratio of the HOA in this study was low (0.06), whereas the H/C ratio
was high (2.21). The O/C ratio of HOA of this study was similar to the updated values of HOA
(0.05 − 0.25) from other studies (Canagaratna et al., 2015).
The regular enhancement of HOA around 7:00 − 9:00, as shown in its diurnal profile (Fig. 10f),
was consistent with the occurrence of morning rush hour traffic in Seoul and the association of
HOA with vehicle emissions. HOA concentration decreased rapidly from 8:00 to 12:00 noon and



remained low in the afternoon, mainly due to dilution associated with rising of BL height. A slow
increase of HOA concentration began at ~16:00 and persisted till the next morning, suggesting
that the shallow BL enhanced the gradual accumulation of the pollutants from vehicle emissions.
However, the correlations of the time series of HOA with gaseous tracers of primary emissions
(i.e., BC, $NO_2$, and CO), were only moderate (Fig. 9a and Table 2), mainly because these pollutants
are emitted not only from vehicular sources, but also from other combustion sources, e.g., biomass
burning. Indeed, the correlations are much stronger between these pollutants and total POA (=
HOA + COA + BBOA) (r = 0.7 – 0.81; Fig. 11 and Table 2).
In this study, major differences were observed between weekdays and weekends for HOA
including other primary species, e.g., BC and POA factors (Fig. S10). For example, the diurnal
pattern of HOA and BC changed significantly between the weekdays and weekends, a general
decrease in the morning rush hour peak over the weekend was observed, likely due to a decrease
in commuting activities because people were more likely to be at home. This weekend effect is a
typical urban features which were observed in Fresno (Young et al., 2016) and Northeastern U.S.
(Zhou et al., 2016b), as well.
*3.3.2.2 Cooking OA (COA)*
COA, as resolved by AMS OA spectra, has been widely reported in urban areas with high
population densities (He et al., 2010;Huang et al., 2010;Mohr et al., 2012;Sun et al., 2011b;Young
et al., 2016;Ge et al., 2012a;Wang et al., 2016;Xu et al., 2014); however no results have yet been
reported from Seoul. In this study, COA was found to account for 20% of the total OA mass, higher
than HOA (Fig. 10k). The diurnal pattern of COA displayed a large evening peak, with a maximum
concentration occurring at 19:00, i.e., dinner time. Elevated COA concentration and larger
fractional contribution to OA mass were observed throughout the night (Figs. 10g, l).
Similar to HOA, the mass spectrum of COA in this study also contained many alkyl fragments,
but to a lesser extent (75.8 % of the total signal in COA spectrum compared to 87.9% of the total
signal in HOA spectrum) (Fig. 10b). However, COA contains significantly larger amounts of
oxygen-containing ions than HOA (e.g., $C_xH_yO_1^+$ = 15.4% vs. 7.6% and $C_xH_yO_2^+$ = 5.1% vs. 2.3%)
(Fig. 10a), and thus had a higher O/C ratio (0.14) and a lower H/C ratio (1.89). The OM/OC ratio
was 1.3 and the H/C ratio is 1.78 for COA. The observed O/C ratio (0.14) of COA in Seoul was at
the lower end of the range (0.14-0.27) of the revised measured O/C ratio of COA in other studies,
e.g., Barcelona (0.27) (Mohr et al., 2012), New York City (NYC) (0.23) (Sun et al., 2011a) and



Fresno (0.14 in 2010 (Ge et al., 2012b) and 0.19 in 2013 (Young et al., 2016)). Previous studies
suggest that $C_3H_3O^+$ (*m/z* 55) and $C_3H_5O^+$ (*m/z* 57) were major fragments of aliphatic acids (e.g.,
linoleic acid and palmitic acid) in cooking oils or animal fat and therefore used these ions as key
tracers for identifying the presence of aerosols from cooking related activities (He et al.,
2004;Adhikary et al., 2010;Mohr et al., 2009;Zhao et al., 2007). In addition, $C_5H_8O^+$ (*m/z* 84) and
$C_6H_{10}O^+$ (*m/z* 98) have been proposed as AMS tracers for COA as well (Ge et al., 2012a;Sun et
al., 2011b). In this study, the time series of COA correlated well with these ions, e.g., $C_3H_3O^+$ (r =
0.86), $C_5H_8O^+$ (r = 0.93), $C_7H_{12}O^+$ (r = 0.89), and $C_6H_{10}O^+$ (r = 0.95) (Fig. 9b and Table 2) and
COA was a major contributor to the signals of $C_5H_8O^+$, $C_6H_{10}O^+$, and $C_7H_{12}O^+$, accounting for
57%, 69%, and 52%, respectively, of their signals (Fig. S6). To show the chemical difference
between COA and other OA factors, Mohr et al. (2012) used the relationships between the fractions
of OA signals at m/z 55 and m/z 57 (i.e., $f55$ and $f57$) or between those of $C_3H_3O^+$ and $C_3H_5O^+$
(i.e., $fC_3H_3O^+$ and $fC_3H_5O^+$) after subtracting the contributions from the oxygenated OA factors
and found that the ratios between m/z 55 ($C_4H_7^+$ + $C_3H_3O^+$ ) and *m/z* 57 ($C_4H_9^+$ + $C_3H_5O^+$ ) in
COA were much higher (2.2 − 2.8) than the ratios (0.9-1.1) in other POAs (e.g., HOA and BBOA).
The COA resolved in Seoul in this study had an *m/z* 55 to *m/z* 57 ratio of 2.2, within the range of
the values for COA reported in Mohr et al. (2012) (Fig. S9). In addition, the ratios between $f55$
and $f57$ for OA in Seoul increased proportionally as the fractional contribution of COA to total
OA increased (Fig. S9b), with a "V" shape indicated by the two edges defined by the COA and
the HOA factors from several urban AMS data sets (Mohr et al., 2012). These observations
together confirm the identification of COA at Seoul.
***3.3.2.3 Biomass burning OA (BBOA)***
Wood combustion was found to be another important POA source (23 %, Fig.10k) in Seoul during
winter, in addition to vehicle and cooking emissions. BBOA is typically prevalent during winter
in locations where wood is used for residential heating (Ge et al., 2012a;Crippa et al., 2013;Zhang
et al., 2015;Young et al., 2016). The mass spectrum of BBOA showed strong signals of oxygenated
ions ($C_xH_yO_1^+$: 27.1% of total BBOA signal and $C_xH_yO_2^+$: 10.6 % of total BBOA signal) and was
more oxidized than HOA and COA (Fig. 10). Among the three POA factors, the O/C ratio of
BBOA was the highest (0.34) and the H/C ratio was the lowest (1.74), similar to the results reported
in several previous studies (e.g., Aiken et al., 2009;Ge et al., 2012a;Mohr et al., 2012). An analysis
of the OA spectra (Fig. 10c) revealed the typical features of BBOA, with dominant peaks at *m/z*





60 (100% being $C_2H_4O_2^+$) and $m/z$ 73 (95% being $C_3H_5O_2^+$), which are known fragments of
levoglucosan and related species (e.g., mannosan and galactosan) (Cubison et al., 2011). Scatter
plots of $f44$ versus $f60$ indicate a higher $f60$ and lower $f44$ (i.e., toward the center of the triangle
area of the biomass burning plumes) as the relative importance of BBOA to the total OA increased
(Fig. S9). The $f44$ and $f60$ of BBOA (0.01 vs. 0.015) in this study were also within the range of
values found for the other ambient BBOA factors or biomass burning aerosols from chamber
studies. HOA and COA, in contrast, had much lower $f60$ values (< 0.01). The time series of BBOA
correlated well with $C_2H_4O_2^+$ (r = 0.85) and $C_3H_5O_2^+$ (r =0.74) (Fig. 9d and Table 2) as well as
other biomass burning tracer species, including potassium (r = 0.63) and BC (r = 0.82). BBOA
also correlated well with nitrogen-containing species, particularly $C_3H_4N^+$ (r = 0.75), $C_2H_4N^+$ (r =
0.70) and $CHN^+$ (r = 0.58), which was consistent with the emissions of nitriles from biomass
burning activities (Simoneit et al., 2003). There was also a strong correlation between the
concentration of polycyclic aromatic hydrocarbons (PAH; r = 0.90) and BBOA, indicating that
biomass burning was a main source of PAH in Seoul during wintertime. Similarly, it was found in
Fresno, California, during wintertime that BBOA correlates well with N-containing ions and PAHs
(Ge et al., 2012a;Young et al., 2016). Given that PAHs are byproducts of incomplete combustion,
many of which are mutagenic and carcinogenic (Dzepina et al., 2007;Hannigan et al., 1998;Marr
et al., 2006), our findings suggest that adverse health effects associated with biomass burning
emissions should be of concern during winter in the Seoul region.
BBOA was the most abundant primary OA in Seoul during this study, account for 39% of the POA
mass and 23% of the total OA mass. However biomass burning is not the main fuel for the
residential heating in Seoul and thus, the BBOA observed in this study must have originated from
other wood burning activities, either locally or regionally. The fact that the observed BBOA was
relatively oxidized (O/C=0.34), with a large contribution from $m/z$ 44 ($f44$ = 5.1%), was consistent
with the observations of more aged BBOA instead of primary woodstove emissions (Crippa et al.,
2013;Zhang et al., 2015;Zhou et al., 2016a). The diurnal profile of BBOA showed an enhancement
at around 10:00 and a background BBOA concentration of ~ 1 $\mu g/m^3$. Given that the polar plot of
BBOA showed high concentrations at both low and high wind speeds (Fig. 7), the sources of
BBOA in Seoul likely include both local and regional wood burning activities. Local wood burning
activities were possibly for the purposes of heating open and public areas (e.g., construction areas,
market), dispose of leaves and woody trash in the city, and heating some residences. Regional




sources of BBOA are possibly from open biomass burning in agricultural area near Seoul (Heo et
al., 2009) and transport from North Korea or further from Russia (Jung et al., 2016).
*3.3.2.4. Semi-volatile and low volatile oxygenated OA (SV-OOA and LV-OOA)*
In addition to the three POA factors, two OOA factors were identified and were found to account
for an average of 41% of the OA mass (Fig. 10k). OOA is ubiquitous and dominant in the
atmosphere (Jimenez et al., 2009;Zhang et al., 2007a) but usually accounts for less than half of the
OA mass observed in urban locations during winter such as NYC, Tokyo, Fresno and Manchester
(Zhang et al., 2007b).
In many cases, OOA can be further separated into a low volatile/more oxygenated OOA (LV-
OOA/MO-OOA) and a semi-volatile/less oxygenated OOA (SV-OOA/LO-OOA), which represent
different degrees of aging and oxidation (Jimenez et al., 2009;Ng et al., 2010, and reference
therein;Setyan et al., 2012;Xu et al., 2015). In this study, two OOA factors, SV-OOA and LV-
OOA were observed to account for 15 and 26% of the total OA mass, respectively (Fig. 10k). As
shown in the triangle plots in Fig. S9, SV-OOA (O/C = 0.56; H/C = 1.90) resides within the region
representing fresher SOA, with a low $f44$, and LV-OOA (O/C = 0.68; H/C = 1.61) was similar to
aged and highly oxidized OA, with a high $f44$. It has been observed that fresh OOA becomes
increasingly more oxidized and less volatile through aging processes in the atmosphere resulting
in LV-OOA and that the evolution of SOA is regarded as a continuum of oxidation. The mass
spectra of both LV-OOA and SV-OOA were very similar to the spectra of OOA factors reported
in other cities (e.g., Hayes et al., 2013;Mohr et al., 2012;Zhang et al., 2014).
Comparisons between the time series of SV-OOA and LV-OOA factors with gaseous species ($NO_2$,
$CO_2$, $O_3$ and $SO_2$), aerosol species, and meteorological parameters further confirmed their
secondary nature. As described in Table 2, SV-OOA and LV-OOA strongly correlated with nitrate
(r = 0.87 and 0.63, respectively) and sulfate (r = 0.71 and 0.80, respectively), whereas the
correlations between POA factors and inorganic species were low (r = 0.09 – 0.41). The Pearson's
correlation coefficient between total OOA (= SV-OOA + LV-OOA) and the sum of secondary
inorganic aerosols ($NO_3$ + $SO_4$ + $NH_4$) was as high as 0.91 (Fig. 11a). These results confirm the
association of SV-OOA and LV-OOA with SOA.
As discussed above, sulfate in Seoul mainly is mainly associated with regional sources, while
nitrate is often formed more locally due to the intense urban emissions of $NO_x$. The better
correlations between SV-OOA and nitrate and between LV-OOA and sulfate (Table 2) suggest



that SV-OOA likely had more local sources whereas LV-OOA likely had more regional sources.
Furthermore, SV-OOA correlated more strongly with methanesulfonic acid (MSA), an SOA
species that tends to be semi-volatile. As shown in Table 2, the correlations of SV-OOA and LV-
OOA with AMS spectral ions for MSA (Ge et al., 2012a), i.e., $CH_3SO_2^+$ (r = 0.90 and 0.53,
respectively) and $CH_2SO_2^+$ (r = 0.83 and 0.53, respectively) corroborated the different natures of
SV-OOA (fresher, more local) and LV-OOA (aged, more regional). The diurnal profiles of SV-
OOA and LV-OOA also reflected the features of local versus regional sources (Figs. 10 i,j). SV-
OOA concentration had a clear peak during mid-morning to afternoon (10:00-11:00, Fig. 10i);
however, LV-OOA concentration was relatively constant throughout the day (Fig. 10j). Finally,
the polar plots of both OOAs showed more dispersed features compared to the POA factors,
especially HOA and COA, but SV-OOA appeared to have a stronger association with local
processes, as its high concentrations tended to be associated with lower wind speed, compared to
LV-OOA  (Fig. 7).
**3.4   Relative Importance of local and regional influences on air quality in Seoul**
**during winter**
In an effort to improve ambient air quality, the Korean government enacted "Special Act on
Seoul Metropolitan Air Quality Improvement" to regulate the concentrations of key pollutants
such as $SO_2$, CO, $NO_2$, $O_3$, PM, and Pb (lead) in 2005. However, till today, Seoul is still facing
poor air quality problems, especially in terms of high concentrations of $PM_{2.5}$ and $O_3$. $PM_{2.5}$ has
been one of the primary concerns due to its detrimental impacts on human health as well as on
visibility. $O_3$ is an important air pollutant itself and can contribute to the secondary formation of
$PM_{2.5}$. Since the development of effective air pollution control policies must rely on knowledge
about the sources, it is important to investigate the major formation processes and emission sources
that contribute to the high PM loadings. Therefore, in this section, we examine how both primary
emissions and secondary formation affect PM loadings in Seoul during winter.
Fig. 12 shows comparisons of the average concentrations of $PM_1$ components as well as other
air pollutants under high and low PM loading conditions depicted on Figs. 2 and 3. The average
concentrations of all aerosol components and OA sources were 1.7 – 8.6 times higher during the
high loading periods compared to the low loading periods (Table 3, Figs. 12e,f). A main reason
appeared to be meteorological conditions. For example, high loading periods were generally
stagnant with low wind speed (0.99 ± 0.7 m/s) (Figs. 13a,b), leading to the accumulation of





pollutants, especially those mainly from local sources. Indeed, among all species, SV-OOA, HOA,
nitrate, and COA showed the highest increases during high loading periods and their average
enhancement in concentrations were 8.6, 5.2, 4.7 and 4.5 times, respectively, of the values during
the low loading periods (Fig. 12e). In addition to accumulate primary pollutants, stable
meteorology condition can also lead to longer atmospheric residence time, which facilitates the
local formation of secondary species such as SV-OOA and nitrate. Furthermore, the relatively high
RH during the high loading periods (71% ± 15; Table 3, Fig. 12a) likely also enhanced the
formation of secondary species such as sulfate and LV-OOA through aqueous-phase processing.
A further evidence for enhanced aqueous-phase processing of secondary aerosol species during
high PM loading periods is shown in Fig. 13; the size distributions of all secondary inorganic
species (nitrate, sulfate and ammonium) were significantly larger during the more polluted periods,
peaking at 500-600 nm in $D_{va}$, compared to the cleaner periods (peaking at 300-400 nm). A
previous study in a US city during winter time also observed that high RH conditions, thus
enhanced aqueous-phase processing, led to increases in the size modes of sulfate, nitrate, and
ammonium (Ge et al., 2012b).

16       The enhancement ratios of the other primary pollutants such as CO, $SO_2$, $NO_2$, BC, and BBOA,

were in the range of 1.2 – 2.5, significantly lower than those of HOA and COA (Fig. 12e),
reflecting the facts that they all had bigger contributions from regional sources compared to HOA
and COA (Fig. 7). Average $O_3$ concentration, on the other hand, showed a substantial decrease (by
~ 70%) during high aerosol loading periods (Fig. 12e and 12f). In addition to enhanced titration
reactions by $NO_x$, another possible reason was reduced photochemical reactions  due to inhibition
of light by high concentration of PM (He et al., 2014).

23       The high loading periods corresponded closely to air masses that are classified as cluster 4 (Fig.

14), which had the shorted trajectories, i.e., slowest wind speeds, as well as the lowest travel height
compared to the other three clusters. The air mass in cluster 4 thus likely held larger amounts of
pollutants and precursors from the ground. In addition, PM in this type of air masses could also be
more oxidized, containing a larger fraction of secondary pollutants,  due to longer residence time
in the atmosphere, therefore was composed of a higher fraction of nitrate.

29       The average aerosol composition during the high loading periods (Fig. 12) was similar to that

for the whole period (Fig. 4), consistent with frequent occurrence of high aerosol pollution
episodes and indicate that these events determined the overall characteristics of $PM_1$ in Seoul





during winter. Given that local emissions were mostly responsible for these pollution events,
controlling the emissions of both primary aerosol particles and precursors for secondary species
from local sources might be an effective way to manage air quality in Seoul during winter time.
Low PM loading periods (average $\pm$ 1$\sigma$ = 12.6 $\pm$ 7.1 $\mu$g m$^{-3}$ for PM$_1$) were commonly
associated with high WS (1.8 $\pm$ 1.1 m/s), low RH (50 %),and long distance transport of air masses
from northern China (such as Inner Mongolia), Mongolia or North Korea (i.e., Cluster 1,2 and 3
from backtrajectory analysis; Fig. 14). Aerosol composition was somewhat different between high
loading and low loading periods. Since strong wind could inhibit the accumulation of local primary
and secondary species while bring in pollutants from upwind sources, the mass fractions of species
influenced by local sources, such as nitrate, SV-OOA, HOA, and COA were lower, whereas those
of regional sources such as sulfate, LV-OOA, BBOA enhanced during low loading periods
compared to more polluted periods (Fig. 12). Although Clusters 1,2 and 3 all represented regional
transport conditions, PM mass concentrations and compositions were somewhat different because
of different origin of air masses. Specifically, Cluster 1 and 2 were almost directly to the north
whereas Cluster 3 was more towards the west (Fig. 14). In comparison, Cluster 1 and 2 had higher
fractions of BBOA but a lower fraction of LV-OOA and sulfate. A possible explanation for this
observation is that northwest area might have more anthropogenic sources than the north area does.
As shown in Fig. 2, PM concentration often changed abruptly with the appearance and
dissipation of a high PM$_1$ event occurring rather quickly (within several hours). The changes were
commonly associated with changes in meteorological conditions, especially wind direction and
speed. Similar trends of sudden changes in air quality have also been observed in other studies in
China, and have been attributed to meteorology (Zhang et al., 2015). For these reasons, it appears
that regional meteorology played an important role in causing high PM pollution conditions in
Seoul.
**4   Conclusions**
Aerosol composition, size distribution, sources, and evolution processes were investigated using
an HR-ToF-AMS and an SMPS in Seoul, Korea, during winter 2015. The average PM$_1$
concentration was 27.5 $\mu$g m$^{-3}$ and the total mass was dominated by organics (44%), followed by
nitrate (24%) and sulfate (10%). Secondary materials (i.e., NO$_3^-$, SO$_4^{2-}$, NH$_4^+$, SV-OOA and LV-
OOA) together accounted for 64% of the PM$_1$ total mass, with the remainder being primary





materials (HOA, COA, BBOA and BC). Cooking, fossil fuel combustion, and wood combustion
were identified as major POA sources in Seoul, contributing an average 59% of total OA mass
during this study.
Meteorological conditions and various emission sources influenced the concentrations,
compositions, size distributions, and properties of aerosol particles in Seoul. High PM pollution
periods tended to build up over a period of 4-5 days and were interleaved with multiple days of
relatively clean periods. High aerosol loading periods were commonly found under relatively
stagnant conditions with low wind speed and aerosol from these periods was characterized with
enhanced fractional contributions of nitrate (27%) and SV-OOA (8%), indicating a strong
influence from local production of secondary aerosol. In contrast, under relatively dynamic
meteorological conditions with higher wind speed, aerosol loading was generally low and $PM_1$
contained larger fractions of species with regional sources, such as sulfate (12%), LV-OOA (20%),
and BBOA (12%). The average O/C ratio of OA was also higher during low loading periods (0.41
vs. 0.35 for high loading periods), indicating the influence of more oxidized OA likely from long
range transport.
Total POA concentration was enhanced by three times during high loading periods. In addition,
enhanced fractional contribution of BBOA was observed during low loading period. In this study,
we found that nearly half of the OA mass was POA and that PAHs in $PM_1$ were mainly from
biomass burning and vehicle emission sources.  These results suggest that it is important to reduce
emissions from combustion sources to improve air quality and protect public health in Seoul during
wintertime. Together, local formation of secondary species (e.g., Nitrate, SV-OOA) during stable
meteorological conditions was also significant for high PM loadings. Thus, the serious pollution
observed in Seoul during winter was caused by combination of factors, including meteorological
conditions, emission by local primary sources, secondary formation as well as transport of air
masses from upwind locations and other unknown factors.
**Acknowledgments**
This work was supported by the Korea Institute of Science and Technology (KIST). QZ
acknowledges the Changjiang Scholars program of the Chinese Ministry of Education.





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





**Tables**
**Table 1.** Average (± 1 standard deviation), minimum and maximum concentrations of the
particulate matter ($PM_1$) species and the total $PM_1$ mass over the whole campaign, and the average
contribution of each of the $PM_1$ species to the total $PM_1$ mass.

| | Average conc. ± one standard deviation ($\mu g\ m^{-3}$) | Minimum conc. ($\mu g\ m^{-3}$) | Maximum conc. ($\mu g\ m^{-3}$) | Fraction of total $PM_1$ (%) | Detection limit (3min/ 6min) ($\mu g\ m^{-3}$) |
|---|---|---|---|---|---|
| Organics | 12.1 ± 8.03 | 1.32 | 53.1 | 44 | 0.03/0.02 |
| Nitrate | 6.55 ± 5.12 | 0.18 | 26.0 | 24 | 0.01/0.01 |
| Sulfate | 2.87 ± 2.00 | 0.38 | 10.7 | 10 | 0.01/0.01 |
| Ammonium | 3.11 ± 2.20 | 0.21 | 10.5 | 12 | 0.02/0.01 |
| Chloride | 0.26 ± 0.32 | 0 | 2.91 | 1 | 0.01/0.01 |
| Black carbon | 2.54 ± 1.66 | 0 | 10.1 | 9 | 0.1/0.05 |
| Total $PM_1$ | 27.5 ± 17.2 | 1.32 | 90.7 | - | 0.04/0.03 |



**Table 2.** Correlation coefficient (Pearson's r) for the linear regressions between organic aerosol
(OA) factors (including the sum of primary factors (primary OA (POA) = hydrocarbon like OA
(HOA) + cooking OA (COA) + biomass burning OA (BBOA)), as well as the sum of the oxidized
factors (oxidized OA (OOA) = semi-volatile OOA (SV-OOA) + low volatile (LV-OOA)), and
various particle- and gas-phase species, and ions.

| r | HOA | COA | BBOA | POA (HOA+ COA+BBOA) | SV-OOA | LV-OOA | OOA (SV-OOA+ LV-OOA) |
|---|---|---|---|---|---|---|---|
| Nitrate | 0.32 | 0.30 | 0.41 | 0.43 | **0.87** | 0.63 | **0.87** |
| Sulfate | 0.25 | 0.17 | 0.09 | 0.23 | **0.71** | **0.80** | **0.88** |
| Ammonium | 0.41 | 0.30 | 0.37 | 0.45 | **0.87** | 0.69 | **0.90** |
| Chloride | **0.80** | 0.30 | 0.50 | 0.68 | 0.50 | 0.13 | 0.36 |
| K (AMS) | 0.59 | 0.60 | 0.63 | **0.77** | 0.61 | 0.22 | 0.47 |
| Primary pollutants | | | | | | | |
| PAH | 0.48 | 0.60 | **0.90** | **0.81** | 0.37 | -0.11 | 0.14 |
| BC | 0.63 | 0.56 | **0.82** | **0.83** | 0.69 | 0.18 | 0.50 |
| CO | 0.52 | 0.54 | 0.62 | **0.74** | 0.61 | 0.29 | 0.51 |
| NO$_2$ | 0.45 | 0.64 | 0.61 | **0.72** | 0.57 | 0.20 | 0.44 |
| AMS tracer ions (*m/z* value) | | | | | | | |
| CO$_2^+$ (44) | 0.32 | 0.38 | 0.41 | 0.47 | **0.84** | **0.75** | **0.92** |
| C$_2$H$_5$N$^+$ (43) | 0.46 | 0.52 | 0.48 | 0.62 | 0.52 | 0.35 | 0.50 |
| C$_2$H$_4$O$_2^+$ (60) | **0.70** | 0.66 | 0.85 | **0.92** | 0.67 | 0.10 | 0.44 |
| C$_3$H$_5$O$_2^+$ (73) | **0.71** | **0.76** | **0.74** | **0.94** | **0.66** | 0.15 | 0.46 |
| C$_3$H$_3$O$^+$ (55) | 0.55 | **0.86** | 0.64 | **0.88** | 0.66 | 0.24 | 0.51 |
| C$_3$H$_7^+$ (43) | **0.91** | 0.69 | 0.63 | **0.96** | 0.46 | 0.02 | 0.27 |
| C$_3$H$_7$N$^+$ (57) | 0.60 | 0.57 | 0.56 | **0.74** | **0.78** | 0.33 | 0.63 |
| C$_4$H$_7^+$ (55) | **0.85** | **0.78** | 0.65 | **0.98** | 0.49 | 0.04 | 0.30 |
| C$_4$H$_9^+$ (43) | **0.95** | 0.62 | 0.61 | **0.94** | 0.42 | 0.00 | 0.24 |
| C$_5$H$_{11}^+$ (57) | **0.96** | 0.59 | 0.60 | **0.92** | 0.41 | -0.01 | 0.23 |
| C$_5$H$_8$O$^+$ (84) | 0.58 | **0.93** | 0.54 | **0.90** | 0.50 | 0.09 | 0.33 |
| C$_6$H$_{10}$O$^+$ (98) | 0.57 | **0.95** | 0.51 | **0.89** | 0.41 | 0.02 | 0.24 |
| C$_7$H$_{12}$O$^+$ (112) | 0.57 | **0.89** | 0.58 | **0.88** | 0.53 | 0.09 | 0.35 |
| C$_9$H$_7^+$ (115) | **0.82** | **0.74** | **0.72** | **0.96** | 0.18 | -0.05 | 0.10 |
| CHN$^+$ (27) | 0.47 | 0.47 | 0.58 | 0.63 | **0.73** | 0.53 | **0.73** |
| CN$^+$ (26) | 0.35 | 0.35 | 0.46 | 0.48 | 0.57 | 0.42 | 0.57 |
| CH$_2$SO$_2^+$ (77) | 0.30 | 0.24 | 0.37 | 0.37 | **0.83** | 0.53 | **0.79** |
| CH$_3$SO$_2^+$ (78) | 0.37 | 0.27 | 0.41 | 0.44 | **0.91** | 0.54 | **0.83** |

BC, black carbon; AMS, aerosol mass spectrometer; PAH, polycyclic aromatic hydrocarbons
Value that are r > 0.7 are boldfaced





**Table 3.** Comparison of aerosol properties and meteorological parameters between the high
loading and low loading periods.

| | High PM loading | Low PM loading |
|---|---|---|
| Average non-refractory submicrometer particulate matter (NR-PM$_1$) mass concentration ($\mu$g m$^{-3}$) (Average ± 1$\sigma$) | 43.6 ± 2.4 | 12.6 ± 7.1 |
| O/C (H/C) ratio* | 0.36 (1.82) | 0.41 (1.75) |
| Gas conc.(CO (ppm) and /NO2/O3/SO2 (ppb)) | 1.2/56/61/7.5 | 0.5/30/18.4/6.4 |
| Temperature (°C) (average ± 1$\sigma$) | 2.5 ± 3.4 | -2.8 ± 4.4 |
| RH (%) (average ± 1$\sigma$) | 71 ± 15 | 50 ± 12 |

*calculated using the improved Canagaratna-ambient method (Canagaratna et al., 2015).
PM, particulate matter; NR-PM$_1$, non-refractory submicrometer particulate matter; RH, relative humidity


1    Figures

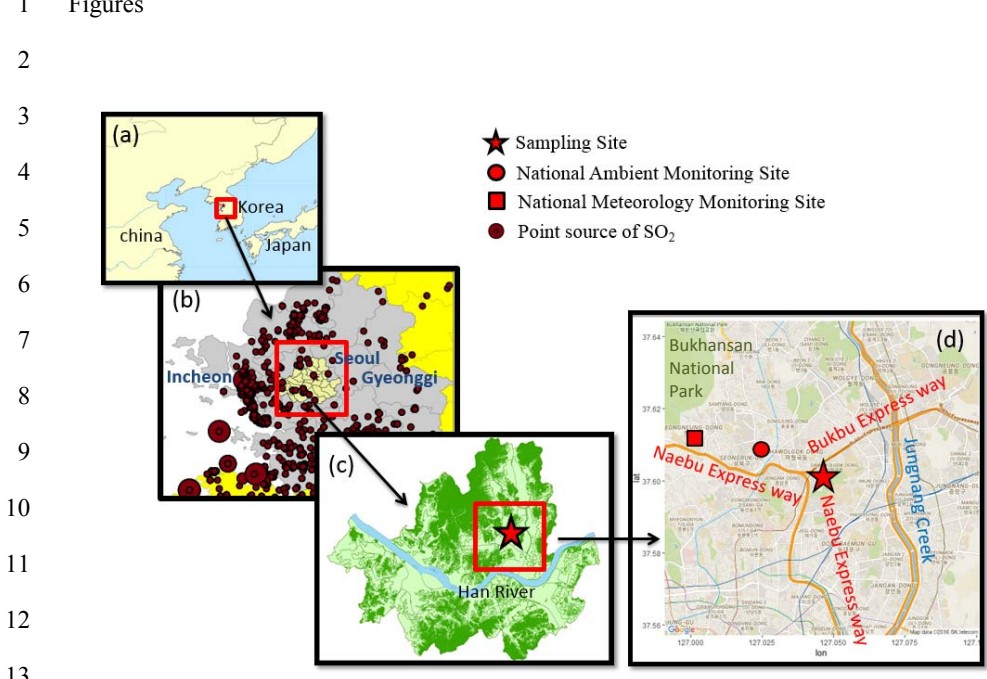

**Figure 1. (a)** The map of Korea, showing the location between China and Japan; **(b)** Seoul (light
yellow) surrounded by other nearby cities including Incheon and Gyeonggi (Grey area) where
industrial facilities are located (west) and agricultural areas are sporadically located surrounding
Seoul. Location of point source of $SO_2$ are indicated by dark red circles (Size: Concentration).
Agricultural areas are sporadically located at East; **(c)** The location of sampling site in Seoul which
is at the north-east of the city center and north of Han river; **(d)** The indications of sampling site.
Next to sampling site, Bukbu express highway is located and residential and commercial areas are
surrounding the sampling site.





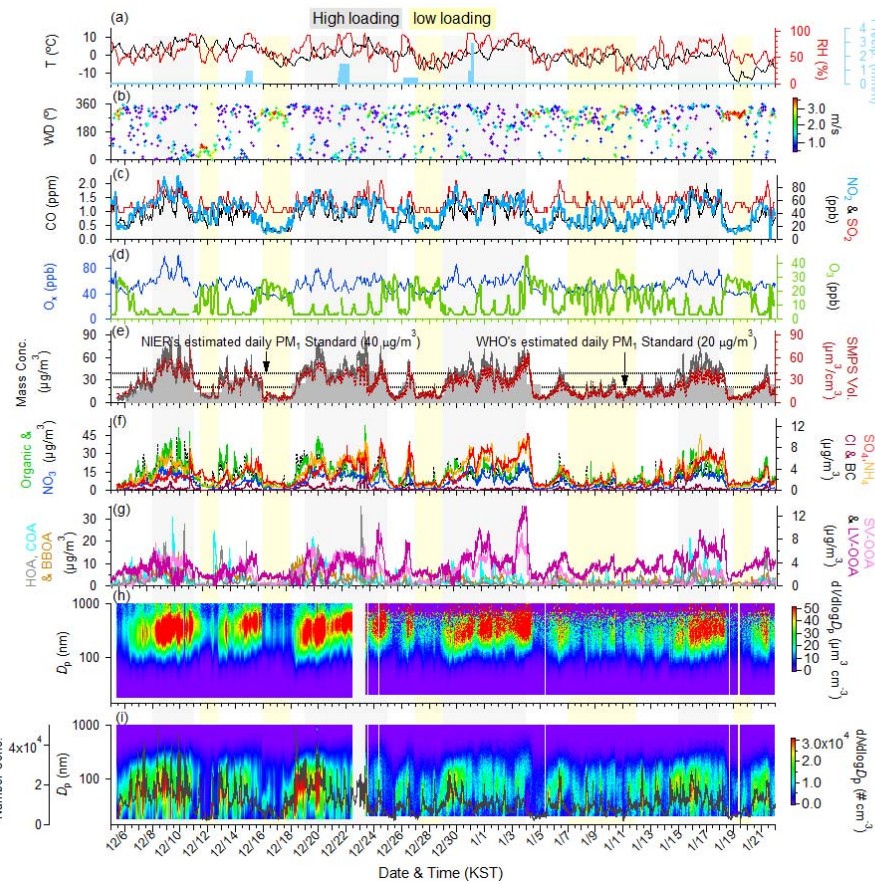

Figure 2. Overview of the temporal variations of submicron aerosols at the Korea Institute of Science and Technology (KIST) in Seoul from December 8, 2015 to January 21, 2016: (a) Time series of ambient air temperature (T), relative humidity (RH), solar radiation (SR), and precipitation (Precip.); (b) Time series of wind direction (WD), with colors showing different wind speeds (WS); (c) Time series of CO, $SO_2$, and $NO_2$; (d) Time series of $O_x$ ($NO_2 + O_3$) and $O_3$; (e) Time series of total particulate matter ($PM_1$), scanning mobility particle sizer (SMPS) volume concentrations Also shown are the 24 h averaged $PM_1$+BC with bars. Estimated NIER's and WHO's daily $PM_1$ standards (40 μg/m$^3$ and 20 μg/m$^3$, respectively) are also shown with dashed line for the comparisons; (f) Time series of the organic aerosols (Org.), nitrate ($NO_3^-$), sulfate ($SO_4^{2-}$), ammonium ($NH_4^+$), chloride ($Cl^-$), and BC; (g) Time series of the total organic aerosol





(OA) of the five factors derived from the positive matrix factorization (PMF) analysis (see section
3.3) **(h,i)** Particle volume and number size distributions by SMPS. Shaded regions indicate the
persistent (> 2days) high loading (gray) and low loading (yellow) periods when daily average conc.
> 30 µg/m$^3$ and <14 µg/m$^3$, respectively.



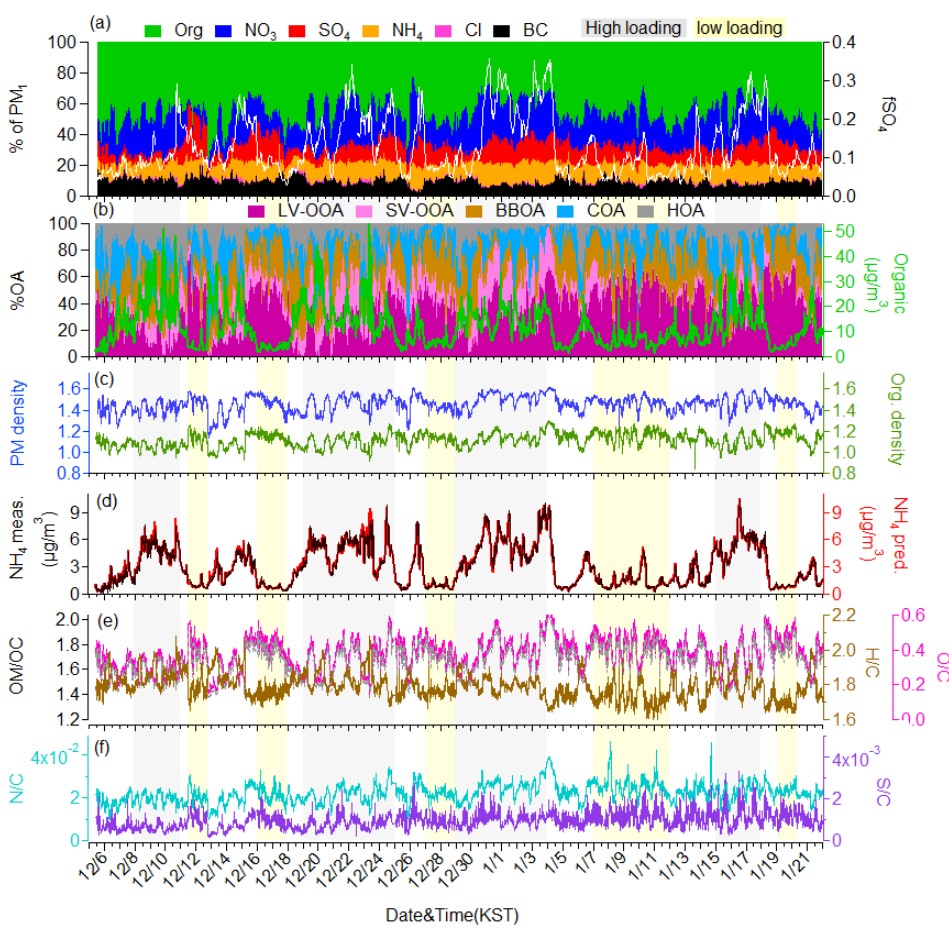

**Figure 3.** Overview of the chemical composition of submicron aerosols at the Korea Institute of

Science and Technology (KIST) in Seoul from December 8, 2015 to January 21, 2016: **(a)** Time

series of the mass fractional contribution of organic aerosols (Org.), nitrate ($NO_3^-$), sulfate ($SO_4^{2-}$

), ammonium ($NH_4^+$), chloride ($Cl^-$), and BC to total $PM_1$ Also shown is the time series of f $SO_4$

(ratio of $SO_4^{2-}/(SO2+ SO_4^{2-})$)with the fraction of $SO_4^{2-}$ to the concentration of ; **(b)** Time series of

the mass fractional contribution to total organic aerosol (OA) of the five factors derived from the

positive matrix factorization (PMF) analysis (see section 3.3), and the time series of the organic

aerosols; **(c)** Time series of the organic density estimate using the method reported in (Kuwata et

al., 2012) with bulk aerosol density using the organic density estimated in this figure; **(d)** Time





series of the measured and predicted $NH_4^+$ concentrations; **(e, f)** organic matter to organic carbon
(OM/OC), oxygen to carbon (O/C), hydrogen to carbon (H/C), nitrogen to carbon (N/C), and sulfur
to carbon (S/C) ratios of OA, where the O/C, H/C and OM/OC elemental ratios were determined
using the updated method (Canagaratna et al., 2015).

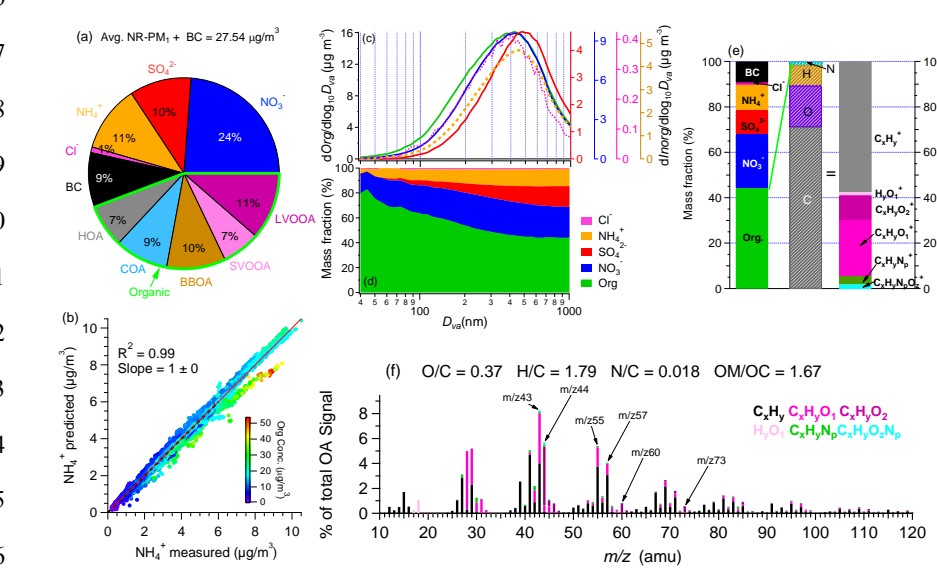

**Figure 4. (a)** Average compositional pie chart of $PM_1$ species (non-refractory-$PM_1$ plus black
carbon (BC)) and each of the OA factors over the whole campaign. The green outline indicates the
fraction of total OA; **(b)** Scatterplot that compares predicted $NH_4^+$ versus measured $NH_4^+$
concentrations. The predicted values were calculated assuming full neutralization of the anions
(e.g., sulfate, nitrate, and chloride). The data points are colored by organic concentrations.; **(c)**
Campaign-averaged size distributions for individual NR-$PM_1$ species; **(d)** Averaged mass
fractional contributions of each NR-$PM_1$ species to the total NR-$PM_1$ mass as a function of size. ;
**(e)** Overview of the average $PM_1$ and OA compositions in Seoul during winter; **(f)** Average high-
resolution mass spectrum of OA colored by the different ion families. The average elemental ratios
for the OA fraction are described.

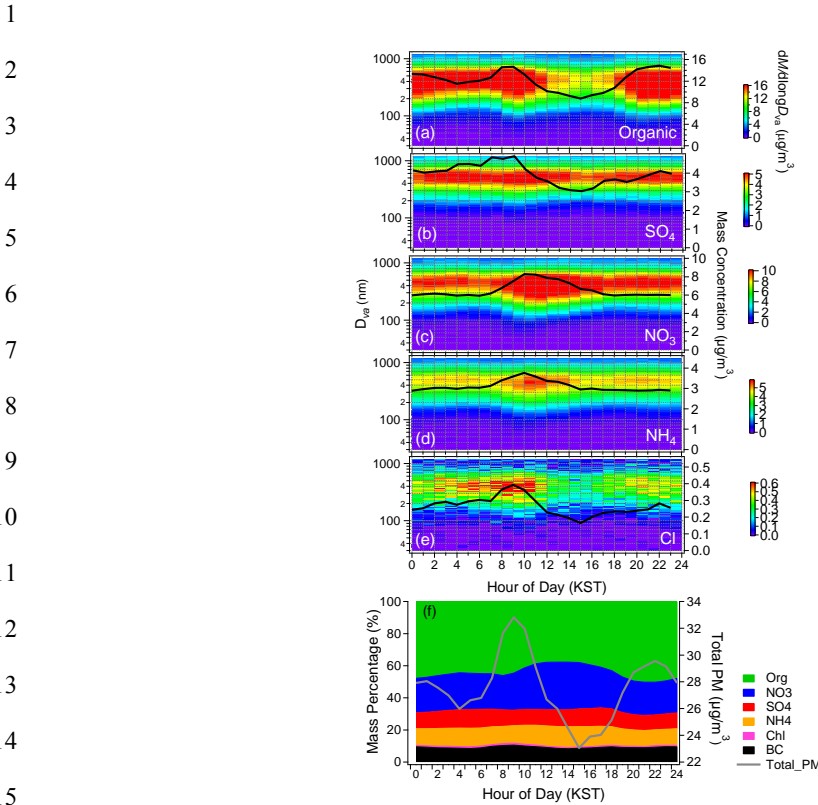

**Figure 5. (a–e)** One-hour averaged diurnal profiles of mass-based size distributions of each of the non-refractory submicrometer particulate matter (NR-PM$_1$) species colored by their mass concentration (left axis; $D_{va}$, vacuum aerodynamic diameter) and average diurnal profiles of each of the PM$_1$ species (right axis); **(f)** Average diurnal mass fractional contribution of each of PM$_1$ species to the total PM$_1$ diurnal mass and the total PM$_1$ mass loading






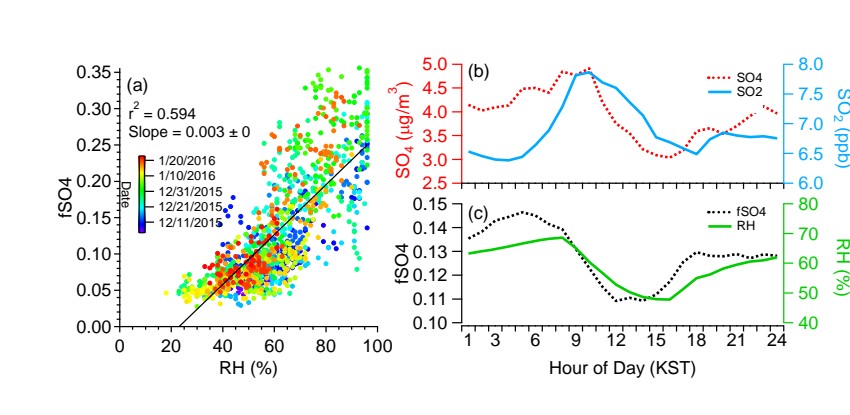

**Figure 6.** **(a)** Scatterplot of the variations of $fSO_4$ ratios as a function of RH **(b)** One-hour averaged
diurnal profiles of $SO_2$ and $SO_4$ and **(c)** One-hour averaged diurnal profiles of $fSO_4$ and RH.


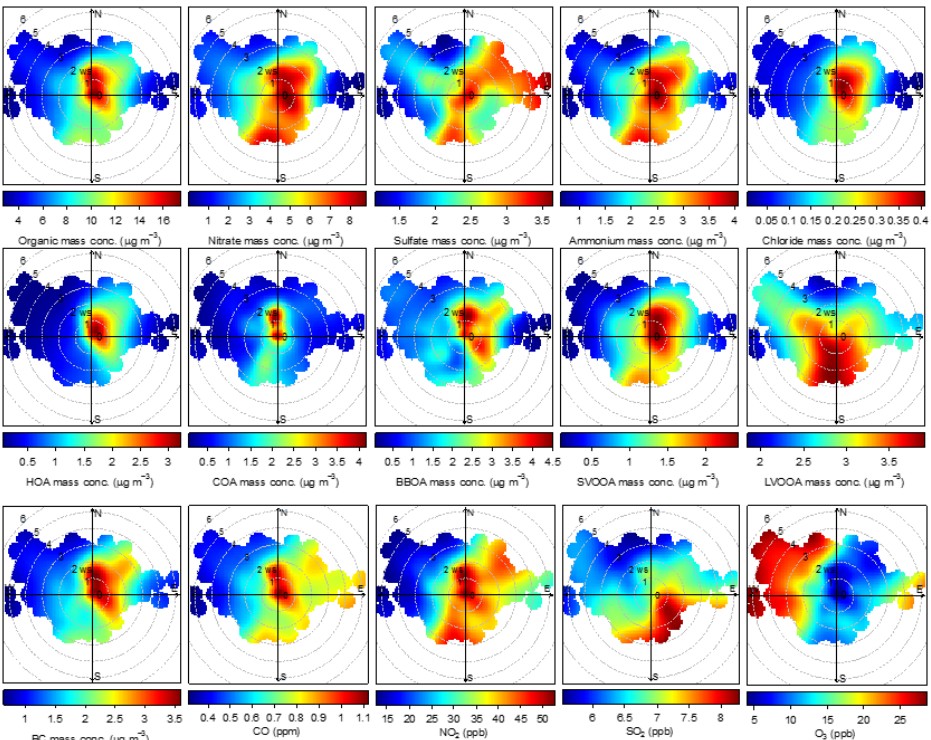

**Figure 7.** Polar plots of hourly averaged PM$_1$ species concentrations (top row), mass
concentrations of the five OA factors identified from PMF analysis (middle row), and the mixing
ratios of various gas phase species as a function of WS and direction.





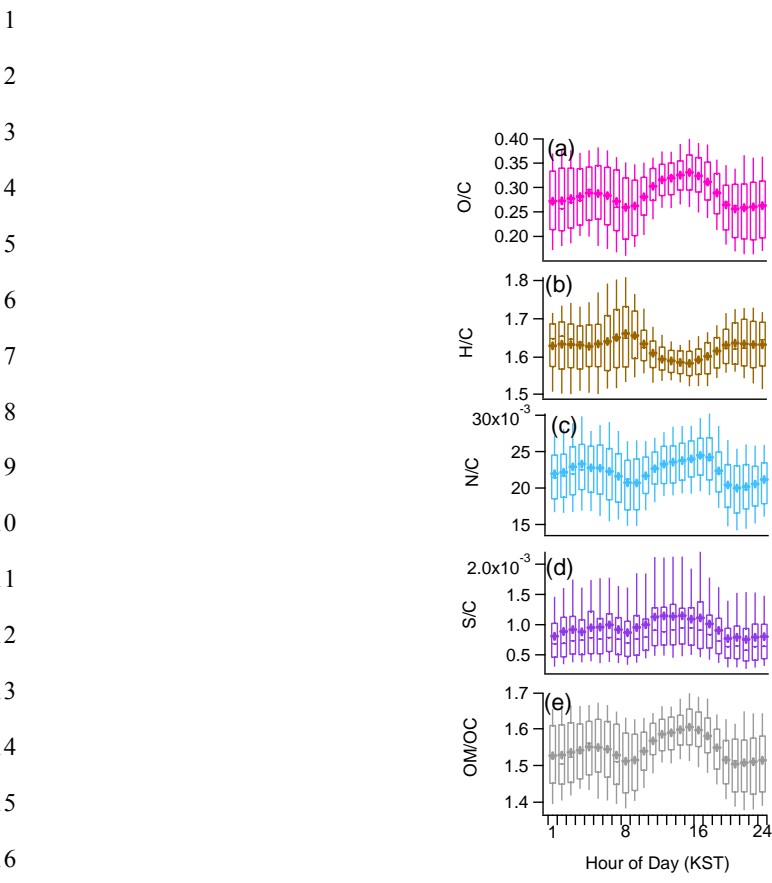

**Figure 8. (a–e)** Average diurnal profiles of the organic matter to organic carbon (OM/OC), oxygen
to carbon (O/C), hydrogen to carbon (H/C), nitrogen to carbon (N/C), and sulfur to carbon (S/C)
ratios of OA, where the O/C, H/C and OM/OC elemental ratios were determined using the updated
method (Canagaratna et al., 2015).







5  **Figure 9.** Overview of the results from PMF analysis including high-resolution mass spectra of

6  the Time series of each of the OA factors and various tracer species.

16





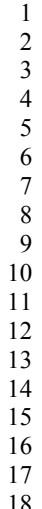

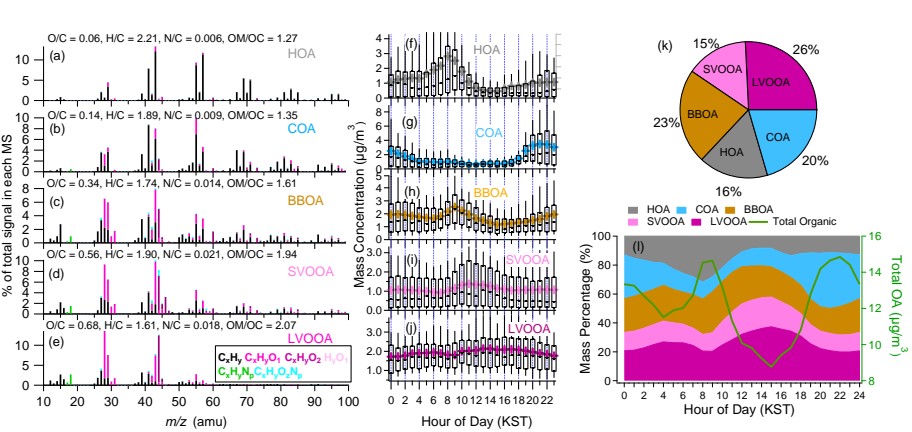

**Figure 10.** Overview of the results from PMF analysis including high-resolution mass spectra of

the **(a)** Hydrocarbon-like organic aerosol (HOA), **(b)** Cooking OA (COA), **(c)** Biomass burning

OA (BBOA), **(d)** Semi volatile oxygenated OA (SV-OOA), and **(e)** Low volatility oxygenated

OA (LV-OOA) colored by different ion families; **(f–i)** Average diurnal profiles of each of the

OA factors (the 90th and 10th percentiles are denoted by the whiskers above and below the boxes,

the 75th and 25th percentiles are denoted by the top and bottom of the boxes, the median values

are denoted by the horizontal line within the box, and the mean values are denoted by the colored

markers); **(k)** Compositional pie chart of the average fractional contribution of each of the OA

factors to the total OA over the campaign; **(l)** Average diurnal mass fractional contribution of

each of the OA factors to the total OA diurnal mass and the total OA mass loading.





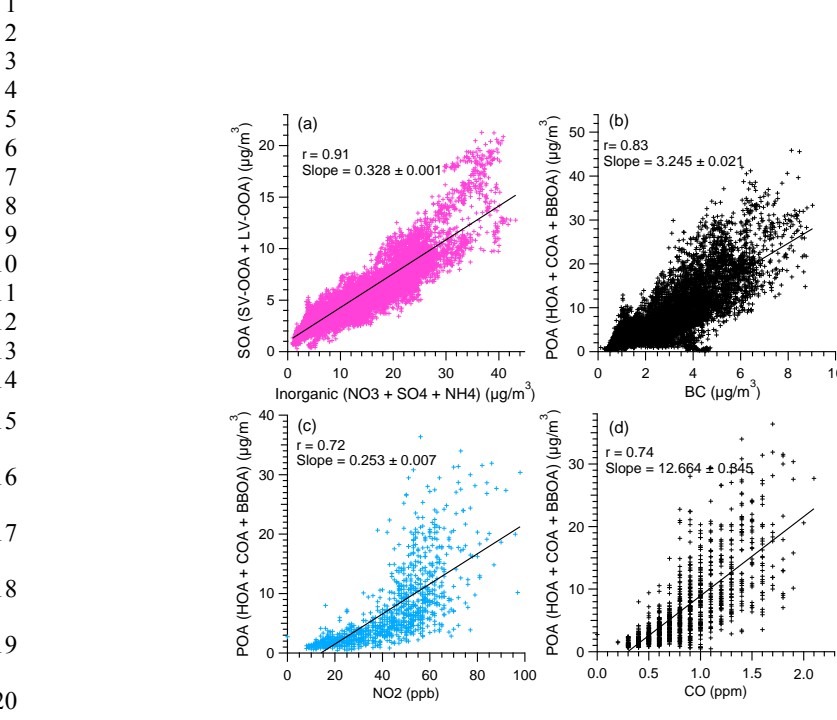

**Figure 11.** Scatterplot between: **(a)** SOA and sum of inorganic (NO3 + SO4 + NH4) **(b)** POA and
BC; **(c)** POA and NO2; and **(d)** POA and CO



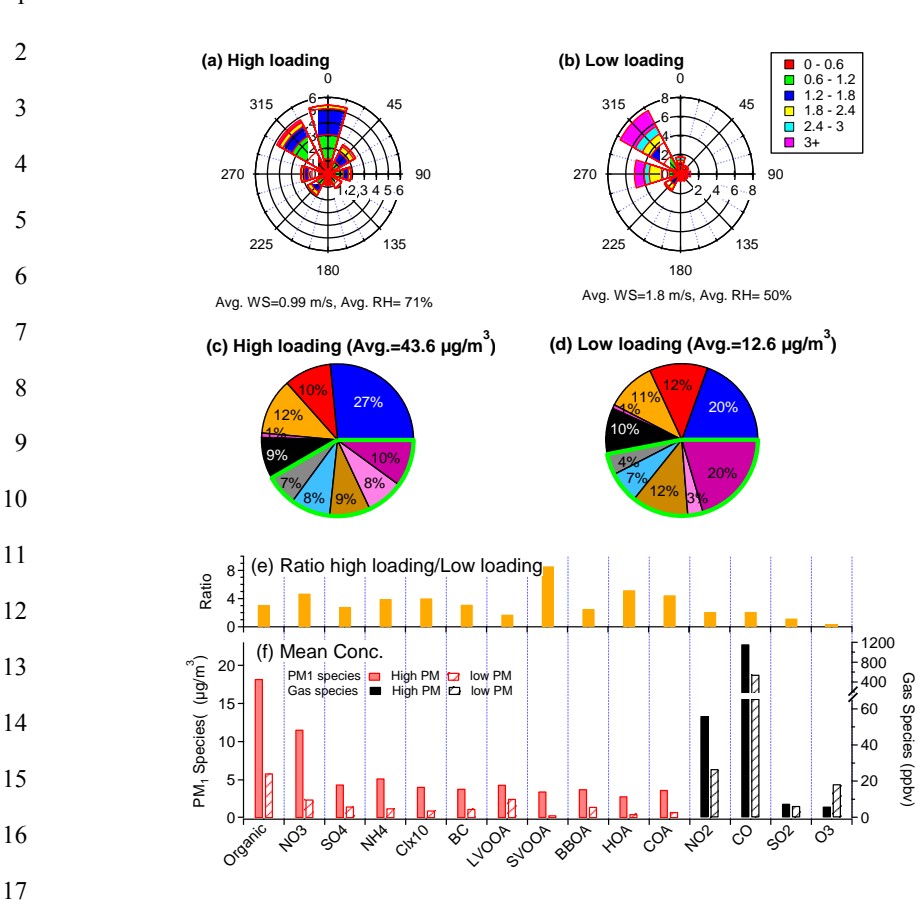

**Figure 12.** Comparisons of averaged properties measured during high particulate matter (PM),
loading and low PM loading periods; **(a,b)** Wind rose plots, colored by wind speed for each
different period; **(c,d)** fractional contributions of each species to the total $PM_1$ (non-refractory-
$PM_1$ plus BC) mass; **(e)** ratios of absolute concentrations of each $PM_1$ and gas species during high
loading and low loading periods; **(f)** Comparisons of averaged absolute concentrations of $PM_1$
species and gaseous pollutants for each different period.

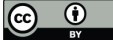



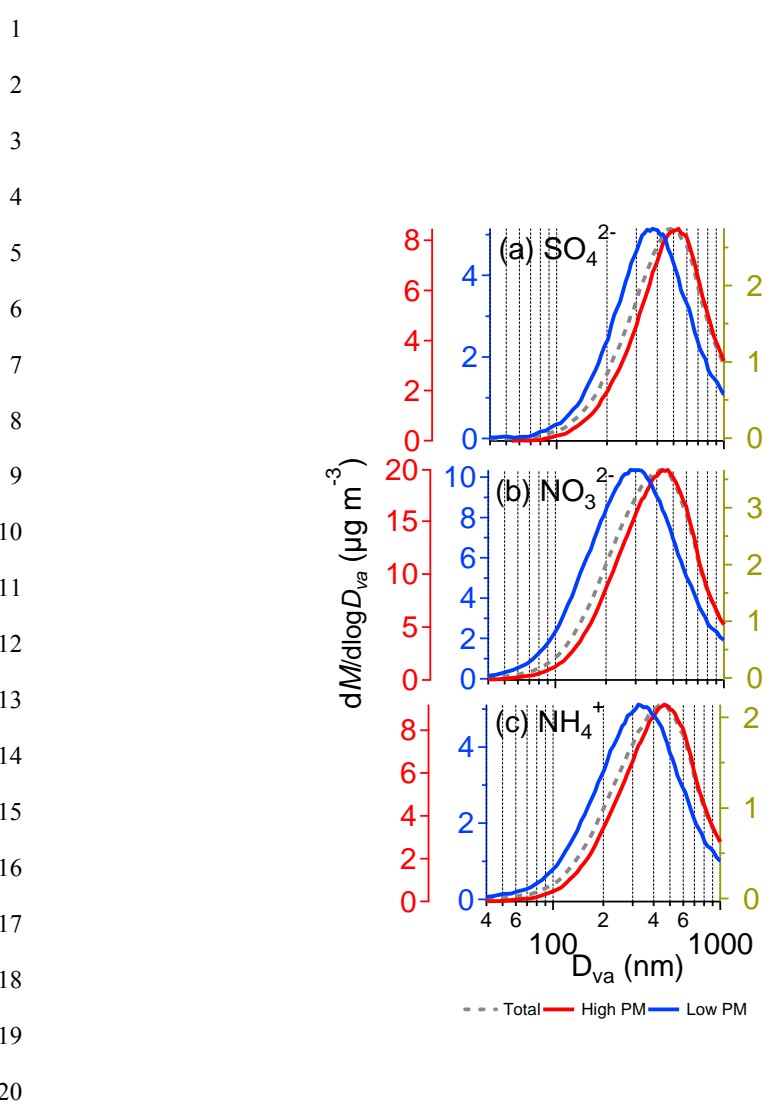

22    **Figure 13.** Averaged mass-based size distributions of **(a)** sulfate, **(b)** nitrate, and **(c)** ammonium

23    during the entire, high PM, and low PM periods as marked in Fig. 2 and 3.

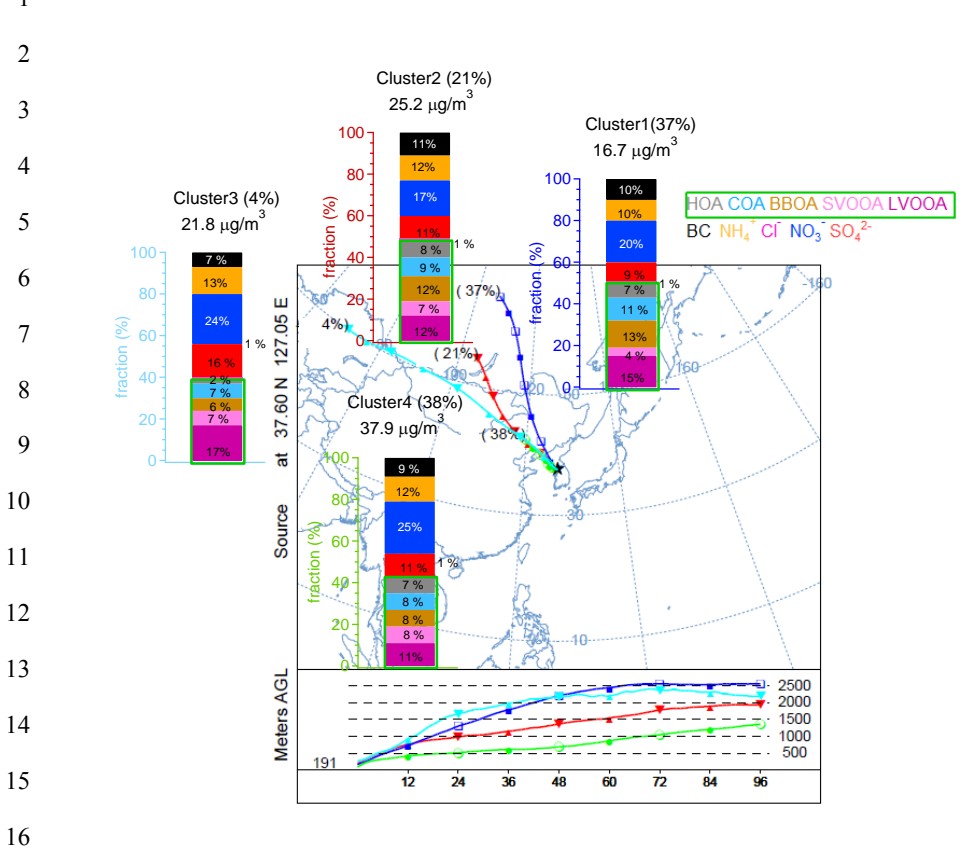

19  **Figure 14.** Averaged compositional bar graph of PM₁ species (non-refractory-PM₁ plus black

20  carbon (BC)) and each of the OA factors in different clusters from four cluster solution at arriving

21  500 m height.

