# Peer review of "Sources and atmospheric processing of wintertime aerosols"

_Atmospheric Chemistry and Physics, 2016_

## Referee Comment (RC1) · Anonymous Referee #1 · 6 Nov 2016

**Reviewer comment**

**Sources and atmospheric processing of wintertime aerosols in Seoul, Korea: Insights from real-time measurements using a high-resolution aerosol mass spectrometer**

*H. Kim et al., Atmos. Chem. Phys. Discuss., doi:10.5194/acp-2016-855, 2016*

**Anonymous reviewer #1**

**General comments**

This manuscript reports results obtained during a field campaign performed at Seoul, South Korea, during 6 weeks in winter 2015/16. The authors deployed an Aerodyne high-resolution time-of-flight aerosol mass spectrometer (hereafter the AMS) and several co-located instruments to measure the particle concentration, chemical composition, and size distribution. The authors performed the usual data treatment, including a source apportionment of organics by positive matrix factorization, a comparison of the different air masses impacting the sampling site with a back trajectory analysis, and a comparison of several selected periods (high vs. low loading).

The authors have a very interesting dataset in the hands, and all the information included in the supplementary material shows that the data treatment has been done very carefully. The manuscript is well written, and the content will be of high interest for the readers of Atmospheric Chemistry and Physics. I warmly recommend the publication of this manuscript after the authors address the following comments.

**Specific comments**

1) Section 2.2 "Measurements": in addition to the five main species, PAHs results are also quickly given later in the Section 3. So I would suggest that the authors mention here that the total PAHs concentrations were also extracted from the organic mass spectra, with eventually some information on the related entries in the fragmentation table.

2) Page 10, lines 19-21: the authors quickly mention that new particle formation events were observed several times during this study. This is an important observation which deserves some additional discussion in the section 3.2 "Size distributions of the main components of $PM_1$" (unless the authors plan to write a separate paper on this topic). In particular, it would be very interesting to use the pToF data of the AMS to identify which chemical species were involved in these new particle growth events.

3) Section 3.1.2 "Diurnal patterns of $PM_1$ composition": I think it would be important to include here a discussion on the diurnal pattern of the wind direction. Indeed, according to Figure S11, the wind direction has a clear diurnal pattern, with a wind coming from the south during the day, and from the west during the night. Therefore, diurnal patterns of the different species discussed in this section seem not only driven by the dynamics of the boundary layer height or photochemistry, but also by a shift in the wind direction.

4) Page 13, lines 27-30: the authors claim that the size distributions of sulfate stayed fairly constant in concentrations and size distribution. Actually, this is not really the case, given that the concentration dropped quite fast between 9h and 14h (Figure 5b). I'm also wondering whether there is a discrepancy between the diurnal pattern of sulfate shown in Figure 5b (strong diurnal pattern, varying between 3 and 5 $\mu g/m^3$) and the hourly averaged size distributions shown in

Figure S4b, where the surface areas of the 24 size distributions look very similar (so no diurnal pattern observed here).

Then, they say that the size distributions of nitrate stayed relatively constant throughout the day as well. This is also not the case, given that the size distributions became much broader between 9h and 15h (Figures 5c and S4a).

5) Page 24, line 17: the authors say that during low loading periods, there was an enhancement of the fractional contribution of BBOA. Actually, according to Figures 12c and 12d, it is mainly LV-OOA which had an important enhancement (from 10 to 20%). In the same time, BBOA increased only from 9 to 12%.

6) Page 24, line 19: Actually, the authors had previously showed the high correlation between PAHs and BBOA, but they did not say anything about the link between PAHs and vehicle emissions. According to Table 2, the correlation between PAHs and HOA is much smaller (r = 0.48, vs. 0.90 between PAHs and BBOA).

7) Figure 14: clusters 1, 2, and 3 seem to come more or less from the same region in the northwest of the sampling site. It would be interesting to include in the supplementary material four maps (one per cluster) with the details of all the individual back trajectories included in each cluster. These graphs are easily generated by the HYSPLIT software and can show us whether a given cluster is the average of similar back trajectories, or whether back trajectories are completely different but give an average cluster coming from the northwest.

**Technical comments**

8) Page 3, lines 23-24: "on air pollution as well"

9) Page 4, lines 16-17: "the mass detection of  non refractory submicrometer particulate matter (NR-PM$_1$)"

10) Page 9, line 16: the authors mention that the ending location of the back trajectory analysis was at an elevation of 500 m. This is obviously not the case in Figure 14, where the ending location is at a much lower altitude than 500 m.

11) Page 12, lines 25-26: "could  be due to the dilution effect"

12) Page 13, lines 28-29: "constant in concentrations and size distribution (Fig. 5b)"

13) Page 16, lines 4-5: the O/C ratios given here do not correspond to those reported in Figure 10(a-e) and Table S1 (neither to the Aiken-ambient method, nor to the improved Canagaratna-ambient method).

14) Page 20, line 29: "sulfate in Seoul  is mainly associated with regional sources"

15) Page 22, line 19: "Average O$_3$_concentration"

16) Page 23, line 5: "low RH (50 %),_and long distance transport"

17) Table 2: if r = 0.70 is the threshold for boldfaced values, please put the following value in bold:
    - C$_2$H$_4$O$_2^+$ (60) vs BBOA

    On the other hand, the following value must not be boldfaced:
    - C$_3$H$_5$O$_2^+$ (73) vs SV-OOA

18) The resolution of several figures (2, 3, 7, 9, S2, S3, S7) is quite poor. Please pay attention to this point during the submission of the final version of the manuscript.

19) Figure 1: the bottom right part of panel (b) is hidden by panel (c). It would be important to shift the different panels, so that we see the entire panel b with the locations of all the point sources of $SO_2$ (currently, we do not see those located on the southeast of the sampling site). BTW, it would be also interesting to mention in section 2 "Experimental methods" where the authors found all the information about the numerous point sources and their concentrations.

20) The title of the supplementary material is not the same as the one of the main manuscript.

21) Figure S1: in the panels (c) and (d), the labels of the left y-axes as well as those of the two colorscales are missing.

22) Figure S4: the legend of the y-axis of the panel (b) should be "$dSO_4$…" instead of "$dNO_3$…".

---

## Referee Comment (RC2) · Anonymous Referee #2 · 8 Nov 2016

Review of

**"Sources and atmospheric processing of winterime aerosols in Seoul, Korea: Insights from real-time measurements using a high-resolution aerosol mass spectrometer"**

By Kim et al.

General comments

The study reports real-time characterization and source apportionment of atmospheric $PM_1$ in Seoul, South Korea during winter 2015. Secondary species, i.e. sulfate, nitrate, ammonium, SV-OOA, and LV-OOA, is found to contribute significantly to the ambient $PM_1$. The PM composition is influenced by meteorological conditions, i.e. temperature and relative humidity. Wind speed and direction are particularly important in characterizing regional and local sources of OA. Locally produced OA, i.e. BBOA, COA, and HOA, contributes majorly to the total OA mass, suggesting the importance of air pollution control in Seoul during winter season.

This study is interesting and important as it is one of the first studies in the region to intensively characterize ambient PM in real-time. The results will be useful for developing or improving air pollution control and policy.

Overall the manuscript falls within the scope of Atmospheric Chemistry and Physics journal. The manuscript is well written and only some revisions are needed. Some statements in the manuscript need to be clarified and/or discussed further. After the following comments are addressed, I recommend the manuscript to be accepted for publication.

Specific comments

Experimental methods:

- PAHs measurements are not described in the methods section, yet the data shows up in discussion (e.g. Table 2). Information about PAH measurements should be added either in the methods, or in the SI and refer to it in the main text.

- For the backtrajectory analysis, what is the air mass estimated to arrive at the location?

Discussion:

- Pg 10 Ln 9: is the severe haze event related to the high loading periods (Fig. 2)?

- Pg 12 Ln 9-12: it could be helpful to add solar radiation data to Fig. 6 to give an idea of when photochemistry possibly occurs at the location. If solar radiation is not available, temperature could give some insights too.

- Pg 21 Ln 6-13: what does the lack of striking diurnal profile of LV-OOA suggest? I think the lack of diurnal variation is related to the regional source of LV-OOA, which has been observed in other areas, e.g. Budisulistiorini et al. (2015, ACP) in USA, Mohr et al. (2012, ACP) in Spain.

- Pg 23 Ln 4-7: it would be good to add short descriptions of what areas/regions are represented by each cluster. The readers may not be familiar with geography in the study location.

Figure 14:

- What does the percentage correspond to? If it is related to the average mass concentration of $PM_1$, the percentages for Cluster 1-3 are incorrect.

- The figure shows that air mass arrive at 191 m agl at the location, whereas on the figure title, it is 500 m (assuming also agl). Is this a typo or they are different terms? The two elevations are different, so please clarify and/or add description about the backtrajectory analysis.

Technical comments

Pg 3 Ln 28: add reference, such as Hennigan et al. (2009, ACP)

Pg 8 Ln 30: insert "respectively" after (SV-OOA).

Pg 10 Ln 7: define NIER

Pg 11 Ln 16: do you mean Fig. 4b?

Pg 12 Ln 4-6: insert "Fig. S10" in the sentence.

Pg 13 Ln 29: it's supposed to be Fig. S4b

Pg 19 Ln 5: *f44* of BBOA is higher than 0.01, I think it's around 0.05. Please check again.

Pg 19 Ln 26-27: BBOA enhances around 9:00 to be more exact.

Pg 20 Ln 21-23: I think $NO_3$ and $SO_4$ instead of $NO_2$ and $SO_2$, respectively are better tracers of oxidized species for comparison with SV-OOA and LV-OOA.

Pg 21 Ln 31: it should be "(Figs. 12a,b)"

Pg 22 Ln 19: space between $O_3$ and concentration.

Pg 22 Ln 21-22: what do you refer by "another possible reason", is it a reason of $O_3$ decreases? If it is so, the sentence needs to be clarified.

Pg 23 Ln 13: delete "compositions". The aerosol compositions are similar. The difference is only concentrations of species at each cluster.

Table 2:

- Correct m/z values for these compounds: $C_4H_9^+$ (57), $C_5H_{11}^+$ (71), $CH_2SO_2^+$ (78), $CH_3SO_2^+$ (79).

- Bold r-value for $C_2H_4O_2^+$ versus BBOA.

Table 3:

- Add "trace gas concentration" on the table caption.

- Subscript the trace gases: e.g. $NO_2$. Also check for this kind of typo elsewhere in the manuscript.

Figure 1c: it is not obvious where the city center is located. Add marker for the city center location.

Figure 2: personally, I think this figure is too crowded. The sub-figures are small and some have many lines (e.g. Fig. 2f, g). Although the lines are colored differently, they are still difficult to differentiate. On, Figure 2i particularly, the line representing average number concentration is almost similar to the gradient color of legend.

Figure 3: be consistent with species on the legend and the caption: e.g. $NO_3$ or $NO_3^-$. Also be consistent in the rest of manuscript.

Figure S4b: y-axis is supposed to be $dSO_4/dLogD_{va}$.

References

Budisulistiorini, S. H., Li, X., Bairai, S. T., Renfro, J., Liu, Y., Liu, Y. J., McKinney, K. A., Martin, S. T., McNeill, V. F., Pye, H. O. T., Nenes, A., Neff, M. E., Stone, E. A., Mueller, S., Knote, C., Shaw, S. L., Zhang, Z., Gold, A., and Surratt, J. D.: Examining the effects of anthropogenic emissions on isoprene-derived secondary organic aerosol formation during the 2013 Southern Oxidant and Aerosol Study (SOAS) at the Look Rock, Tennessee ground site, Atmos. Chem. Phys., 15, 8871-8888, doi:10.5194/acp-15-8871-2015, 2015.

Hennigan, C. J., Bergin, M. H., Russell, A. G., Nenes, A., and Weber, R. J.: Gas/particle partitioning of water-soluble organic aerosol in Atlanta, Atmos. Chem. Phys., 9, 3613–3628, doi:10.5194/acp-9-3613-2009, 2009.

Mohr, C., DeCarlo, P. F., Heringa, M. F., Chirico, R., Slowik, J. G., Richter, R., Reche, C., Alastuey, A., Querol, X., Seco, R., Peñuelas, J., Jiménez, J. L., Crippa, M., Zimmermann, R., Baltensperger, U., and Prévôt, A. S. H.: Identification and quantification of organic aerosol from cooking and other sources in Barcelona using aerosol mass spectrometer data, Atmos. Chem. Phys., 12, 1649-1665, doi:10.5194/acp-12-1649-2012, 2012.

---

## Referee Comment (RC3) · Anonymous Referee #3 · 13 Nov 2016

This manuscript presents a comprehensive measurement study at Seoul during wintertime using a suit of on-line instruments including a HR-ToF-AMS. It is found that the mass concentration of submicron aerosol was high and exceeded the national air quality standard. The chemical species showed evidently diurnal variation suggesting the local and/or regional sources which mainly derived by local weather condition or regional weather systems. The study also found important primary sources and secondary formation pathway for organic aerosol which is important for mitigation for the government. The manuscript is general well done and the data processing is reasonable and thoughtful. The topic of this study is also fitted in the topic of this special issue of ACPD. I recommend it is accepted after a minor revision as followed.

1. The elemental ratio of S/C is generally not used in the reported data because the correcting factor of S/C for ambient data was not well done.

2. Please clarify how PAH was determined? Is it based on default fragmentation or W-mode data?

3. Page11, line 16-18: does any reference to support this point?

line 25-28: please cite this reference:
Xu, J., Zhang, Q., Chen, M., Ge, X., Ren, J., and Qin, D.: Chemical composition, sources, and processes of urban aerosols during summertime in northwest China: insights from high-resolution aerosol mass spectrometry, Atmos. Chem. Phys., 14, 12593-12611, 10.5194/acp-14-12593-2014, 2014.

4. Figure 9: please remove "high-resolution mass spectra of" in the caption.

5. Figure 10: add "+" in each category family.

---

## Author Comment (AC1) · 29 Dec 2016

**Responses to Anonymous Referee #1**

We thank the reviewer for the insightful and valuable comments. Our specific responses are addressed below and colored by blue. Changes made to the manuscript are in quotation marks.

**General comments**
This manuscript reports results obtained during a field campaign performed at Seoul, South Korea, during 6 weeks in winter 2015/16. The authors deployed an Aerodyne high-resolution time-of-flight aerosol mass spectrometer (hereafter the AMS) and several co-located instruments to measure the particle concentration, chemical composition, and size distribution. The authors performed the usual data treatment, including a source apportionment of organics by positive matrix factorization, a comparison of the different air masses impacting the sampling site with a back trajectory analysis, and a comparison of several selected periods (high vs. low loading). The authors have a very interesting dataset in the hands, and all the information included in the supplementary material shows that the data treatment has been done very carefully. The manuscript is well written, and the content will be of high interest for the readers of Atmospheric Chemistry and Physics. I warmly recommend the publication of this manuscript after the authors address the following comments.

**Specific comments**
1) Section 2.2 "Measurements": in addition to the five main species, PAHs results are also quickly given later in the Section 3. So I would suggest that the authors mention here that the total PAHs concentrations were also extracted from the organic mass spectra, with eventually some information on the related entries in the fragmentation table.

Following the suggestion by the reviewer, we have added discussions on PAH estimation on Page 7: "Furthermore, the total concentrations of particle-bound polycyclic aromatic hydrocarbons (PAHs) were estimated using the method described in Dzepina et al. (2007). However, instead of apportioning the unit mass resolution (UMR) spectra, PAH-related ions were determined via fitting the high-resolution mass spectra (W-mode) (Xu et al., 2014). In addition, a RIE of 1.35 with respect to nitrate was applied to calculate mass concentrations of PAHs from AMS data (Dzepina et al., 2007)

2) Page 10, lines 19-21: the authors quickly mention that new particle formation events were observed several times during this study. This is an important observation which deserves some additional discussion in the section 3.2 "Size distributions of the main components of PM1" (unless the authors plan to write a separate paper on this topic). In particular, it would be very interesting to use the pToF data of the AMS to identify which chemical species were involved in these new particle growth events.

Thanks for the suggestions. We are indeed interested in understanding what chemical processes were involved in the new particle growth events in Korea and are currently performing detailed analysis using data both from this study and from a spring campaign when new particle events were also observed. We will report our findings in a future publication.

3) Section 3.1.2 "Diurnal patterns of PM1 composition": I think it would be important to include here a discussion on the diurnal pattern of the wind direction. Indeed, according to Figure S11, the wind direction has a clear diurnal pattern, with a wind coming from the south during the day, and from the west during the night. Therefore, diurnal patterns of the different species discussed in this section seem not only driven by the dynamics of the boundary layer height or photochemistry, but also by a shift in the wind direction.

We mentioned the effect of diurnal variation of wind in the text where sulfate diurnal pattern is discussed: "nighttime transport of air mass from industrial facilities located on the west and southwest outskirts of Seoul (Fig. 1b) might be responsible." In response to the review comment, we have now made it clear in the sentence: "for which nighttime transport of air mass from industrial facilities located on the west (Fig. S11) and southwest outskirts of Seoul (Fig. 1b) might be responsible." For other compounds such as nitrate, since they were mainly associated with local sources, we didn't discuss the effect of wind.

4) Page 13, lines 27-30: the authors claim that the size distributions of sulfate stayed fairly constant in concentrations and size distribution. Actually, this is not really the case, given that the concentration dropped quite fast between 9h and 14h (Figure 5b).
I'm also wondering whether there is a discrepancy between the diurnal pattern of sulfate shown in Figure 5b (strong diurnal pattern, varying between 3 and 5 µg/m3) and the hourly averaged size distributions shown in Figure S4b, where the surface areas of the 24 size distributions look very similar (so no diurnal pattern observed here).

The reviewer was right that the average concentration of sulfate did vary somewhat but sulfate size distribution was indeed relatively constant. For the clarifications, the sentence has been revised as follows: "The size distributions of sulfate showed a prevalent droplet accumulation mode ($D_{va}$ = 500 nm) that stayed fairly constant compared to the one of nitrate (Fig. S4b)."

As for reviewer's comment on a discrepancy between the diurnal patterns of sulfate shown in Figure 5b and Figure S4b, the sulfate concentration in the diurnal profile ranges from 2.5 to 3.3 (The figure has been corrected) which is narrower than the one of nitrate (5.9-8.3). As shown in the graph below, the size distribution of sulfate was similar between the times of maximum and minimum concentrations comparing to the one of nitrate, although there are differences at the left hand side of size distribution curves.

[Figure]

Then, they say that the size distributions of nitrate stayed relatively constant throughout the day as well. This is also not the case, given that the size distributions became much broader between 9h and 15h (Figures 5c and S4a).

This part also has been changed to as follows: "the size distributions of nitrate became broader between 9:00 and 15:00 with significant changes in concentrations (Fig. 5c, Fig. S4a)."

5) Page 24, line 17: the authors say that during low loading periods, there was an enhancement of the fractional contribution of BBOA. Actually, according to Figures 12c and 12d, it is mainly LV-OOA which had an important enhancement (from 10 to 20%). In the same time, BBOA increased only from 9 to 12%.

Thanks for the comments. We agree with the reviewer that the highest enhancement occurred with LV-OOA whereas the enhancement of BBOA was smaller. Nevertheless, we consider an enhancement from 9% to 12 % is significant and worthwhile to mention. In response to the reviewer's comment, we have revised the text to report the actual fractional numbers: "In addition, an enhancement of the fractional contributions of BBOA from the high loading periods (9%) was observed during low loading periods (12%)."

6) Page 24, line 19: Actually, the authors had previously showed the high correlation between PAHs and BBOA, but they did not say anything about the link between PAHs and vehicle emissions. According to Table 2, the correlation between PAHs and HOA is much smaller ($r = 0.48$, vs. 0.90 between PAHs and BBOA).

We have revised the text to make this point clear: "and that PAHs in $PM_1$ were mainly from biomass burning ($r = 0.90$)"

7) Figure 14: clusters 1, 2, and 3 seem to come more or less from the same region in the northwest of the sampling site. It would be interesting to include in the supplementary material four maps (one per cluster) with the details of all the individual back trajectories included in each cluster. These graphs are easily generated by the HYSPLIT software and can show us whether a given cluster is the average of similar back trajectories, or whether back trajectories are completely different but give an average cluster coming from the northwest.

Individual back trajectories in each cluster are now shown in Fig. S12. Also in section 2.4 of the main text, the following sentence has been added: "Individual back trajectories in each cluster are shown in Fig. S12".

**Technical comments**
8) Page 3, lines 23-24: "on air pollution, as well"
Thanks, the sentence has been corrected accordingly.

9) Page 4, lines 16-17: "the mass detection of submicron non refractory submicrometer particulate matter (NR-PM1)"
Done as suggested

10) Page 9, line 16: the authors mention that the ending location of the back trajectory analysis was at an elevation of 500 m. This is obviously not the case in Figure 14, where the ending location is at a much lower altitude than 500 m.

Thanks, it is not 500 m but the half of the mixing height. Therefore, the corrected sentence in the text now reads:
"The trajectories were released at half of the mixing height at the KIST (latitude: 37.60N; longitude: 127.05E) and the average arriving height for the back trajectories for this study was approximately 191 m"

11) Page 12, lines 25-26: "could to be due to the dilution effect"
The sentence has been corrected accordingly.

12) Page 13, lines 28-29: "constant in concentrations and size distribution (Fig. S5b)"
Thanks, it has been corrected to "Fig. S4b"

13) Page 16, lines 4-5: the O/C ratios given here do not correspond to those reported in Figure 10(a-e) and Table S1 (neither to the Aiken-ambient method, nor to the improved Canagaratna-ambient method).
Corrected

14) Page 20, line 29: "sulfate in Seoul mainly is mainly associated with regional sources"
The sentence has been corrected accordingly.

15) Page 22, line 19: "Average O3_concentration"
This has been corrected accordingly.

16) Page 23, line 5: "low RH (50 %),_and long distance transport"
This has been corrected accordingly.

17) Table 2: if r = 0.70 is the threshold for boldfaced values, please put the following value in bold:
- C2H4O2+ (60) vs BBOA

On the other hand, the following value must not be boldfaced:
- C3H5O2+ (73) vs SV-OOA
All have been corrected accordingly.

18) The resolution of several figures (2, 3, 7, 9, S2, S3, S7) is quite poor. Please pay attention to this point during the submission of the final version of the manuscript.
We will include high resolution figures in the revised manuscript.

19) Figure 1: the bottom right part of panel (b) is hidden by panel (c). It would be important to shift the different panels, so that we see the entire panel b with the locations of all the point sources of SO2 (currently, we do not see those located on the southeast of the sampling site). BTW, it would be also interesting to mention in section 2 "Experimental methods" where the authors found all the information about the numerous point sources and their concentrations.
The figure has been amended in the revised manuscript not to hide panel (b). Also this has been addressed in the text and figure legend (concentration). This sentence in the revised manuscript

now reads: "there are a number of industrial facilities that are significant anthropogenic sources for $SO_x$ and the highest emission rate range was measured at 4856-17222 ton/year." Unfortunately, other point sources are not available.

20) The title of the supplementary material is not the same as the one of the main manuscript. Thanks, the title of the supplementary materials has been changed according to the main manuscript.

21) Figure S1: in the panels (c) and (d), the labels of the left y-axes as well as those of the two colorscales are missing.
Thanks, the figure has been corrected accordingly in the revised manuscript.

22) Figure S4: the legend of the y-axis of the panel (b) should be "dSO4…" instead of "dNO3…".
Corrected

References

Dzepina, K., Arey, J., Marr, L. C., Worsnop, D. R., Salcedo, D., Zhang, Q., Onasch, T. B., Molina, L. T., Molina, M. J., and Jimenez, J. L.: Detection of particle-phase polycyclic aromatic hydrocarbons in Mexico City using an aerosol mass spectrometer, Int J Mass Spectrom, 263, 152-170, 2007.

Xu, J., Zhang, Q., Chen, M., Ge, X., Ren, J., and Qin, D.: Chemical composition, sources, and processes of urban aerosols during summertime in northwest China: insights from high-resolution aerosol mass spectrometry, Atmospheric Chemistry and Physics, 14, 12593-12611, 10.5194/acp-14-12593-2014, 2014.

---

## Author Comment (AC2) · 29 Dec 2016

We thank the reviewer for the insightful and valuable comments. Our specific responses are addressed below and colored by blue. Changes made to the manuscript are in quotation marks.

General comments
The study reports real-time characterization and source apportionment of atmospheric PM1 in Seoul, South Korea during winter 2015. Secondary species, i.e. sulfate, nitrate, ammonium, SV-OOA, and LV-OOA, is found to contribute significantly to the ambient PM1. The PM composition is influenced by meteorological conditions, i.e. temperature and relative humidity. Wind speed and direction are particularly important in characterizing regional and local sources of OA. Locally produced OA, i.e. BBOA, COA, and HOA, contributes majorly to the total OA mass, suggesting the importance of air pollution control in Seoul during winter season.
This study is interesting and important as it is one of the first studies in the region to intensively characterize ambient PM in real-time. The results will be useful for developing or improving air pollution control and policy.
Overall the manuscript falls within the scope of Atmospheric Chemistry and Physics journal. The manuscript is well written and only some revisions are needed. Some statements in the manuscript need to be clarified and/or discussed further. After the following comments are addressed, I recommend the manuscript to be accepted for publication.

Specific comments
Experimental methods:
• PAHs measurements are not described in the methods section, yet the data shows up in discussion (e.g. Table 2). Information about PAH measurements should be added either in the methods, or in the SI and refer to it in the main text.

Thanks, as suggested by the reviewer, relevant discussions on PAH estimation has been added at Page 6, line 4-5 as follows: "Furthermore, the total concentrations of particle-bound polycyclic aromatic hydrocarbons (PAHs) were estimated using the method described in Dzepina et al. (2007). However, instead of apportioning the unit mass resolution (UMR) spectra, PAH-related ions were determined via fitting the high-resolution mass spectra (W-mode) (Xu et al., 2014). In addition, a RIE of 1.35 with respect to nitrate was applied to calculate mass concentrations of PAHs from AMS data (Dzepina et al., 2007)"

• For the backtrajectory analysis, what is the air mass estimated to arrive at the location?
For a given trajectory, air mass arrived at the location represents one that passed over different area along the trajectory.
For clarifications, the following sentence has been added at Page 23 line4-6: "In addition, since air masses in each cluster were expected to have passed over regions indicated by the corresponding trajectories, investigating the composition and masses of aerosol in each cluster can shed lights on how various upwind areas influence air quality at the measurement site"

Relating to the comments on Pg 23 Ln 4-7 below, a short description of areas covered by the air mass trajectories is also provided.

Discussion:
• Pg 10 Ln 9: is the severe haze event related to the high loading periods (Fig. 2)?

Yes, the severe haze event occurred during one of the high loading periods (as marked on Fig.2). For clarification, we have revised the text to read, "Although severe haze with high $PM_1$ concentration close to 90 µg m$^{-3}$ was observed several times".

• Pg 12 Ln 9-12: it could be helpful to add solar radiation data to Fig. 6 to give an idea of when photochemistry possibly occurs at the location. If solar radiation is not available, temperature could give some insights too.

The closest available solar radiation measurement data are from 20km away from the measurement site, however solar radiation is likely to be fairly homogeneous over a large area. Therefore, as suggested by reviewer, diurnal pattern of solar radiation has been added to Fig. 6 and data source has been discussed at figure caption as "The solar radiation measurement site is located at 20km away from the measurement site."

• Pg 21 Ln 6-13: what does the lack of striking diurnal profile of LV-OOA suggest? I think the lack of diurnal variation is related to the regional source of LV-OOA, which has been observed in other areas, e.g. Budisulistiorini et al. (2015, ACP) in USA, Mohr et al. (2012, ACP) in Spain.

Yes, we also think that the relatively constant diurnal pattern of LV-OOA suggests regional source of this aerosol component. For the clarification, this sentence has been revised as follow and suggested referenced has been added: "however, LV-OOA concentration was relatively constant throughout the day, suggesting regional sources of this aerosol component (Fig. 10j). Similar observations were also reported in other areas such as North America (e.g., Budisulistiorini et al., 2015;Sun et al., 2011b;Woody et al., 2016;Zhou et al., 2016b;Zhang et al., 2005b), Europe (e.g., Mohr et al., 2012;Young et al., 2015), and Asia (e.g., Huang et al., 2010;Jiang et al., 2015;Wang et al., 2016)"

• Pg 23 Ln 4-7: it would be good to add short descriptions of what areas/regions are represented by each cluster. The readers may not be familiar with geography in the study location.

The regions were briefly mentioned in the text. In response to the reviewer comments, we have added more details about the area and the text now reads: "All three Clusters appeared to originate from Russia, however there were some differences among clusters. For example, cluster 1 passed over Mongolia and North Korea whereas Clusters 2 and 3 passed over China. Furthermore Cluster 3 was composed of the longest trajectories."

Figure 14:
• What does the percentage correspond to? If it is related to the average mass concentration of PM1, the percentages for Cluster 1-3 are incorrect.

The percentage values on the bars are the average fractional contributions of each species to the average $PM_1$ (= NR-$PM_1$ + BC) mass concentration in different clusters. We suspect that the reviewer's comment was related to the total values in Clusters 1-3 not being 100%. That was because of rounding of the values.

• The figure shows that air mass arrive at 191 m agl at the location, whereas on the figure title, it is 500 m (assuming also agl). Is this a typo or they are different terms? The two elevations are different, so please clarify and/or add description about the backtrajectory analysis.
Thanks, it was typo. The arriving height was not 500 m, but the half of the mixing height calculated by the HYSPLIT program. We have revised the sentence and the text now read: "The trajectories were released at half of the mixing height at the KIST (latitude: 37.60N; longitude: 127.05E) and the average arriving height for the back trajectories for this study was approximately 191 m"

Technical comments
Pg 3 Ln 28: add reference, such as Hennigan et al. (2009, ACP)
The reference has been added.

Pg 8 Ln 30: insert "respectively" after (SV-OOA).
The sentence has been corrected.

Pg 10 Ln 7: define NIER
The full name of NIER, National Institute of Environmental Research has been added.

Pg 11 Ln 16: do you mean Fig. 4b?
Thanks, it has been corrected to Fig. 4b.

Pg 12 Ln 4-6: insert "Fig. S10" in the sentence.
Done as suggested

Pg 13 Ln 29: it's supposed to be Fig. S4b
Thanks, it has been corrected to "Fig. S4"

Pg 19 Ln 5: *f44* of BBOA is higher than 0.01, I think it's around 0.05. Please check again.
Thanks, it has been corrected to 0.05.

Pg 19 Ln 26-27: BBOA enhances around 9:00 to be more exact.
Thanks, it has been corrected to 9:00.

Pg 20 Ln 21-23: I think NO3 and SO4 instead of NO2 and SO2, respectively are better tracers of oxidized species for comparison with SV-OOA and LV-OOA.
We agree that NO3 and SO4 are better tracers for SV-OOA and LV-OOA and this point was discussed at the line 24 in the submitted version. For clarification, we removed the lists of gaseous species and added the following sentence to discuss the good correlation of NO3 and SO4 with SV-OOA and LV-OOA: "Comparisons between the time series of SV-OOA and LV-OOA with gaseous species, aerosol species, and meteorological parameters further confirmed their secondary nature. As shown in Table 2, SV-OOA and LV-OOA strongly correlated with nitrate (r = 0.87 and 0.63, respectively) and sulfate (r = 0.71 and 0.80, respectively), whereas the correlations between POA factors and the inorganic aerosol species were low (r = 0.09 – 0.41)."

Pg 21 Ln 31: it should be "(Figs. 12a,b)"

Thanks, it has been corrected to Figs. 12a,b

Pg 22 Ln 19: space between O3 and concentration.
Done.

Pg 22 Ln 21-22: what do you refer by "another possible reason", is it a reason of O3 decreases? If it is so, the sentence needs to be clarified.
We indeed refer it as "another possible reason for O3 decrease". For clarification, the sentence has been changed to "another possible reason for $O_3$ decrease was reduced photochemical reactions due to inhibition of light by high concentration of PM {He, 2014 #416}"

Pg 23 Ln 13: delete "compositions". The aerosol compositions are similar. The difference is only concentrations of species at each cluster.
In fact, aerosol compositions are indeed fairly different among clusters. As shown in Figure 2 and –Figure 12, the average fractions of nitrate (27 vs. 20 %), SV-OOA (8 vs 3 %), HOA (7 vs 4%) and COA (8 vs 7%) decreased from high to low loading periods whereas the fractions of BBOA (9 vs 12 %), LV-OOA (10 vs 20 %), and sulfate (10 vs 12%) all enhanced. For clarifications, specific fractions are now mentioned in the text and the paragraph reads: "Aerosol composition was somewhat different between the high loading and the low loading periods. Since strong wind could inhibit the accumulation of local primary and secondary species while bring in pollutants from upwind sources, the mass fractions of species influenced more strongly by local sources, such as nitrate (27 vs 20 %), SV-OOA (8 vs 3 %), HOA (7 vs 4 %), and COA (8 vs 7 %) were lower during low loading periods compared to more polluted periods, whereas those of regional sources such as sulfate (10 vs 12 %), LV-OOA (10 vs 20 %), BBOA (9 vs 12 %) were enhanced (Fig. 12)."

Table 2:
• Correct m/z values for these compounds: C4H9+ (57), C5H11+ (71), CH2SO2+ (78), CH3SO2+ (79).
• Bold r-value for C2H4O2+ versus BBOA.
Thanks, it has been corrected.

Table 3:
• Add "trace gas concentration" on the table caption.
Done as suggested.

• Subscript the trace gases: e.g. NO2. Also check for this kind of typo elsewhere in the manuscript.
Thanks, it has been corrected in the Table and also check throughout the paper.

Figure 1c: it is not obvious where the city center is located. Add marker for the city center location.
Thanks, we intended to mention that the sampling site is located at the north east of the center of Seoul as shown at the Figure. For the clarification, the Figure caption has been revised to "Center of Seoul" instead of city center.

Figure 2: personally, I think this figure is too crowded. The sub-figures are small and some have many lines (e.g. Fig. 2f, g). Although the lines are colored differently, they are still difficult to differentiate. On, Figure 2i particularly, the line representing average number concentration is almost similar to the gradient color of legend.

As the reviewer suggested, Figure 2 has been modified. Hope that this figure looks better for the interpretation.

Figure 3: be consistent with species on the legend and the caption: e.g. NO3 or NO3-. Also be consistent in the rest of manuscript.
Thanks, it has been revised.

Figure S4b: y-axis is supposed to be dSO4/dLogDva.
Thanks, it has been revised.

References

[revised manuscript text omitted]

---

## Author Comment (AC3) · 29 Dec 2016

**Responses to Anonymous Referee #3**

We thank the reviewer for the insightful and valuable comments. Our specific responses are addressed below and colored by blue. Changes made to the manuscript are in quotation marks.

This manuscript presents a comprehensive measurement study at Seoul during wintertime using a suit of on-line instruments including a HR-ToF-AMS. It is found that the mass concentration of submicron aerosol was high and exceeded the national air quality standard. The chemical species showed evidently diurnal variation suggesting the local and/or regional sources which mainly derived by local weather condition or regional weather systems. The study also found important primary sources and secondary formation pathway for organic aerosol which is important for mitigation for the government. The manuscript is general well done and the data processing is reasonable and thoughtful. The topic of this study is also fitted in the topic of this special issue of ACPD. I recommend it is accepted after a minor revision as followed.

1. The elemental ratio of S/C is generally not used in the reported data because the correcting factor of S/C for ambient data was not well done.

S-containing organic ions (i.e., $C_xH_yO_zS_q^+$; $x > 0$; $y \geq 0$; $z \geq 0$, and $q > 1$) were clearly determined and properly fitted. It is true that no correction factor is available for the AMS measured S/C ratio, the S/C data determined from analyzing the HR-AMS data can nevertheless provide useful information about the temporal and diurnal variations of the S/C ratio in organic aerosol. For clarification, we have added the following sentence at the last paragraph of the section 2.3.1: "Note that there could be biases in the S/C ratios since no correction factor is available."

2. Please clarify how PAH was determined? Is it based on default fragmentation or W-mode data?

Thanks, as suggested by the reviewer, relevant discussions on PAH estimation has been added at Page 6, line 4-5 as follows: "Furthermore, the total concentrations of particle-bound polycyclic aromatic hydrocarbons (PAHs) were estimated using the method described in Dzepina et al. (2007). However, instead of apportioning the unit mass resolution (UMR) spectra, PAH-related ions were determined via fitting the high-resolution mass spectra (W-mode) (Xu et al., 2014). In addition, a RIE of 1.35 with respect to nitrate was applied to calculate mass concentrations of PAHs from AMS data (Dzepina et al., 2007)"

3. Page11, line 16-18: does any reference to support this point?

A study conducted by Ge et al. (2012) reported similar observations as ours. We now cite this reference in the revised text.

Ge, X. L., Zhang, Q., Sun, Y. L., Ruehl, C. R., and Setyan, A.: Effect of aqueous-phase processing on aerosol chemistry and size distributions in Fresno, California, during wintertime, Environ. Chem., 9, 221-235, 10.1071/en11168, 2012

line 25-28: please cite this reference:
Xu, J., Zhang, Q., Chen, M., Ge, X., Ren, J., and Qin, D.: Chemical composition, sources, and processes of urban aerosols during summertime in northwest China: insights from high-resolution aerosol mass spectrometry, Atmos. Chem. Phys., 14, 12593-12611, 10.5194/acp-14-12593-2014, 2014.
This reference has been added.

4. Figure 9: please remove "high-resolution mass spectra of" in the caption.
Thanks, it has been corrected.

5. Figure 10: add "+" in each category family.

Thanks, it has been corrected.

References

Dzepina, K., Arey, J., Marr, L. C., Worsnop, D. R., Salcedo, D., Zhang, Q., Onasch, T. B., Molina, L. T., Molina, M. J., and Jimenez, J. L.: Detection of particle-phase polycyclic aromatic hydrocarbons in Mexico City using an aerosol mass spectrometer, Int J Mass Spectrom, 263, 152-170, 2007.

Xu, J., Zhang, Q., Chen, M., Ge, X., Ren, J., and Qin, D.: Chemical composition, sources, and processes of urban aerosols during summertime in northwest China: insights from high-resolution aerosol mass spectrometry, Atmospheric Chemistry and Physics, 14, 12593-12611, 10.5194/acp-14-12593-2014, 2014.